# Unraveling the epigenetic code: human kidney DNA methylation and chromatin dynamics in renal disease development

Yu Yan[1,2,3,4], Hongbo Liu [1,2,3,4], Amin Abedini [1,2,3,4], Xin Sheng [1,2,3,4], Matthew Palmer[4,5], Hongzhe Li[4,6] & Katalin Susztak [1,2,3,4] ✉

Epigenetic changes may fill a critical gap in our understanding of kidney disease development, as they not only reflect metabolic changes but are also preserved and transmitted during cell division. We conducted a genome-wide cytosine methylation analysis of 399 human kidney samples, along with single-nuclear open chromatin analysis on over 60,000 cells from 14 subjects, including controls, and diabetes and hypertension attributed chronic kidney disease (CKD) patients. We identified and validated differentially methylated positions associated with disease states, and discovered that nearly 30% of these alterations were influenced by underlying genetic variations, including variants known to be associated with kidney disease in genome-wide association studies. We also identified regions showing both methylation and open chromatin changes. These changes in methylation and open chromatin significantly associated gene expression changes, most notably those playing role in metabolism and expressed in proximal tubules. Our study further demonstrated that methylation risk scores (MRS) can improve disease state annotation and prediction of kidney disease development. Collectively, our results suggest a causal relationship between epigenetic changes and kidney disease pathogenesis, thereby providing potential pathways for the development of novel risk stratification methods.

Chronic kidney disease (CKD) ranks as the tenth leading cause of death globally, accounting for approximately one million deaths each year[1]. CKD is a complex disease resulting from gene-environment interactions. Kidney dysfunction, primarily estimated as low glomerular filtration rate (eGFR), has a strong heritable component, with genetic variants potentially accounting for 30–50% of eGFR variability in the general population[2–4]. Most CKD cases in the US arising as a consequence of diabetes, hypertension, aging, ischemic, or toxic insults[5].

Genome-wide association studies (GWASs) have identified over 800 loci where variants are associated with kidney function[4,6–9]. Despite the remarkable success of GWAS, it remains challenging to interpret this information since more than 90% of identified variants reside in the noncoding regions of the genome[10]. Moreover, the identified variants only explain a small fraction of heritability, with factors accounting for the remaining heritability yet to be discovered[11].

Multiple lines of evidence suggest that epigenetic modifications could account for some of the missing heritability of kidney

[1]Renal, Electrolyte, and Hypertension Division, Department of Medicine, University of Pennsylvania, Perelman School of Medicine, Philadelphia, PA 19014, USA. [2]Institute for Diabetes, Obesity, and Metabolism, University of Pennsylvania, Perelman School of Medicine, Philadelphia, PA 19014, USA. [3]Department of Genetics, University of Pennsylvania, Perelman School of Medicine, Philadelphia, PA 19014, USA. [4]Kidney Innovation Center, University of Pennsylvania, Perelman School of Medicine, Philadelphia, PA 19014, USA. [5]Department of Epidemiology and Biostatistics, Perelman School of Medicine, Philadelphia, PA 19014, USA. [6]Department of Pathology, Perelman School of Medicine, Philadelphia, PA 19014, USA. ✉e-mail: ksusztak@pennmedicine.upenn.edu

disease[6,12–16]. Studies indicate that adverse intrauterine conditions, such as calorie restriction or hyperglycemia, contribute to the development of hypertension and kidney disease later in life[17–19]. Additionally, periods of hyperglycemia can hasten kidney function decline in patients with diabetes even after decades of strict glycemic control, a phenomenon known as "metabolic memory"[20,21]. Epigenome-modifying enzymes, which use substrates such as acetyl and methyl groups for post-translational modification of histones or DNA, are highly sensitive to fluctuations in metabolite levels[22–24].

Several studies already explored epigenetic changes in patients with kidney disease. Cytosine methylation changes in blood samples have been analyzed in several large cohorts, including the Pima, CKDGen, DCCT (Diabetes Control and Complications Trial), German CKD Cohort, and in the Chronic Renal Insufficiency Cohort (CRIC)[14–16,25–30]. These studies identified a robust association between methylation of specific loci and kidney function. Using statistical modeling, the DCCT study even suggested that certain epigenetic alterations mediate the hyperglycemia-induced kidney disease development[31]. However, these previous studies have several limitations, including small sample sizes, not differentiating between methylation changes driven by genotypes and those driven by environmental factors, and not linking methylation changes observed in bulk samples to single-cell level differences.

## Table 1 | Clinical and histopathological characteristics of the human kidney samples

| Characteristics (n = 399) | Result |
|---|---|
| eGFR, median (IQR) (range), ml/min per 1.73 m² | 71.2 (24.1) (3.7–134.9) |
| Age, mean (SD), year | 60.2 (13.5) |
| Sex, M, n % | 254 (63.7) |
| Race, Black, n % | 61 (15.3) |
| Diabetes, n % | 136 (34.1) |
| Hypertension, n % | 271 (67.9) |
| BMI, median (IQR), kg/m², (n = 370) | 29.4 (26.0, 34.4) |
| HgbA1c, median (IQR), %, (n = 57) | 6.8 (5.9, 11.1) |
| Serum glucose, median (IQR), mg/dl, (n = 272) | 121.0 (97.0, 151.0) |
| Systolic BP, mean (SD), mmHg | 134.9 (18.8) |
| Diastolic BP, mean (SD), mmHg | 76.5 (11.9) |
| Serum albumin, median (IQR), g/dl, (n = 168) | 4.0 (3.6, 4.3) |
| Pathology | |
| Interstitial fibrosis, median (IQR), % | 5 (2, 10) (0–100) |
| Interstitial lymphocytic infiltrate >2 +, % | 19.8 |
| Interstitial plasmacytic infiltrate present, % | 28.7 |
| Interstitial eosinophilic infiltrate >2 +, % | 1.0 |
| Tubular atrophy, median (IQR), % | 5 (2, 10) |
| Acute tubular injury, median (IQR) (range), % | 0 (0, 2) (0–50) |
| Tubules reabsorption present, % | 23.6 |
| Global glomerulosclerosis, median (IQR), % | 5.7 (2.4, 13.1) |
| Glomerular wall thickening >2 +, % | 2.3 |
| Glomerular hypoperfused >2 +, % | 9.6 |
| Glomerular mesangial matrix >2 +, % | 6.6 |
| Glomerular mesangial cellularity >2 +, % | 6.1 |
| Glomerular KW nodules present, % | 2.5 |
| Glomerular pericapsular fibrosis >2 +, % | 9.0 |
| Vessel medial thickening present, % | 8.2 |
| Vessel intimal fibrosis >2 +, % | 44.9 |
| Vessel arteriolar hyalinosis >2 +, % | 8.5 |

eGFR estimated glomerular filtration rate, M male, BMI body mass index, HgbA1c high sensitivity hemoglobin A1C, IQR interquartile range.

The development of droplet-based encapsulation and barcoding has enabled the analysis of thousands or even millions of single cells and the genome-wide analysis of open chromatin and gene expression changes. Initial analysis of a limited number of samples has allowed for the identification of cell-type-specific open chromatin changes and potential key transcription factors specific to each cell type[32,33]. Analysis of control and disease samples would also enable precise cell-type-specific characterization of epigenetic changes in disease state, but sample number and the availability of human kidneys remains limited. Combining cell-type-specific analysis with bulk profiling of large cohorts could provide comprehensive and complementary information.

In this study, we aimed to address the limitations of previous research by conducting an extensive epigenome-wide analysis of a larger cohort of 399 human kidney tissue samples from controls, diabetic, hypertensive, and chronic kidney disease (CKD) patients, and by employing single-cell open chromatin analysis of over 60,000 cells obtained from 14 subjects. By combining genetic and epigenetic studies we aimed to understand the role of genetic variations driving epigenetic changes and using genotype information for causal inference analysis. Our objectives were to identify differentially methylated sites associated with kidney disease, examine their impact on gene expression, and assess the potential of methylation risk scores (MRS) for disease prediction and diagnosis. By combining genetic information, bulk profiling, and cell type-specific analysis, we aimed to provide comprehensive and complementary information on the role of epigenetic alterations in the pathogenesis of kidney disease.

## Results

### Characteristics of study samples

Our goal was to uncover robust epigenetic changes associated with common manifestations of kidney disease, such as low kidney function (eGFR) and fibrosis. Diabetes and hypertension associated kidney diseases exhibit overlapping clinical manifestations, and combined account for over 75% of all CKD and end-stage renal disease (ESRD) cases in the US[34,35]. As the epigenome is cell-type specific, we analyzed changes in microdissected human kidney tubule samples to reduce the contribution of cell heterogeneity[36]. We collected 399 kidney samples from healthy, diabetic, hypertensive, and diabetic and hypertensive CKD subjects (Table 1). The mean age was 60.2 years. Approximately one-third of the participants had diabetes, and nearly 68% had hypertension. The mean eGFR, calculated by the CKD-EPI formula, was 71.2 ml/min/1.73m², ranging from 3.7 to 134.9 ml/min/1.73m². Approximately one-third (n = 123) of the subjects had an eGFR less than 60 ml/min/1.7 2m², meeting the classic definition of CKD.

We also assessed histological changes in biopsy samples using an unbiased scoring system, evaluated by a pathologist who was blinded to the clinical parameters. The median tubulointerstitial fibrosis (hereinafter referred to as fibrosis) ranged from 0 to 100% with 129 samples with greater than 10% fibrosis. Fibrosis and eGFR showed a significant negative correlation (Pearson correlation coefficient r = −0.46, P < 2.2 ×10⁻¹⁶; Supplementary Fig. 1a). To examine the relatedness of the different clinical and histological variables, we conducted clustering (Supplementary Fig. 1b). Clinical and histological variables separated and formed independent clusters. Variables from independent clusters used in the analysis to prevent model overfitting. In our study, 49 (12%) subjects met the standard definition of DKD, which includes histological changes and reduced eGFR, while 94 (23%) patients had hypertensive CKD.

### Epigenome-wide association analysis (EWAS) identified methylation changes associated with kidney structure and function

In this study, we quantified DNA methylation at over 850,000 CpG sites in 399 kidney tubule samples using Illumina Human Methylation

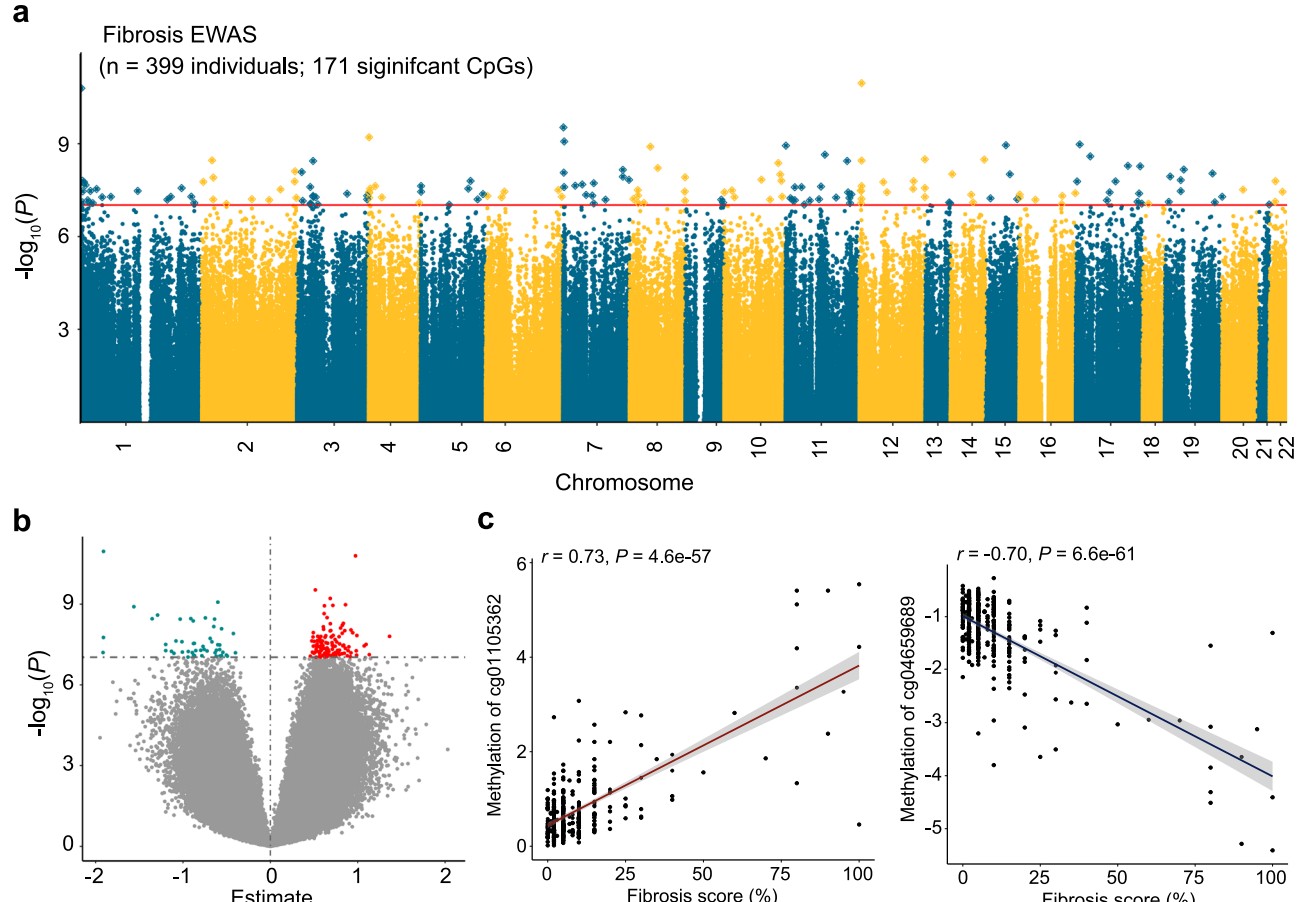

**Fig. 1 | Epigenome-wide association analysis (EWAS) identified methylation changes associated with kidney disease severity. a** Manhattan plot of fibrosis EWAS in 399 human kidney samples. The x-axis represents the chromosomal location of the CpG probes and the y-axis is the -log$_{10}$(*P* value with Bacon-correction) of fibrosis and methylation association. The epigenome-wide significance level (two-sided $P < 9.42 \times 10^{-8}$) is indicated by the red line and significant CpGs are highlighted as rectangles. **b** Volcano plot showing the association between fibrosis and methylation changes. The x-axis represents the effect size (from the linear regression) of each CpG probe with fibrosis and the y-axis indicates the strength of the association (-log$_{10}$(*P* value with Bacon-correction)). Each dot corresponds to one probe, with red dots representing hypermethylated probes and cyan dots representing hypomethylated probes that are associated with higher fibrosis. **c** Correlation between methylation levels and fibrosis score at cg04659689 and cg01105362. Each data dot represents one kidney sample. The x-axis shows the percent kidney fibrosis (0–100%), the y-axis represents the methylation level (as M-value), and the shaded area indicates the 95% confidence interval for the correlations. The r represents Pearson's correlation coefficient and *P* indicates the strength of the association (two-sided).

EPIC arrays. After rigorous data processing and quality control filtering, we investigated the association between fibrosis and DNA methylation levels at 701,519 CpG sites. To account for technical effects and the bimodal distribution of the methylation measurements, we employed linear mixed-effects models, regressing methylation (used as M values) on bisulfite conversion control, mean intensity of measurements, sample plate, and lymphocytic infiltration. Next, we performed a linear regression analysis to assess the association between normalized fibrosis score and residualized methylation, while controlling for potential confounders such as age, sex, race, hypertension, and diabetes status.

We identified 171 CpG sites where cytosine methylation levels correlated with the degree of fibrosis (differentially methylated position: DMP) (Fig. 1a; Supplementary Data 1) at an epigenome-wide significance level ($P < 9.42 \times 10^{-8}$) after correcting inflation. Of the 171 fibrosis-DMPs, 49 (28.7%) exhibited lower methylation in fibrotic samples (Fig. 1b). The most significant signal was observed at cg18566594 (on chromosome 12; Bacon corrected $P = 1.10 \times 10^{-11}$), which was in the promoter region of CCND2-AS1 gene (Supplementary Fig. 2). Methylation levels at the top fibrosis DMPs, cg04659689 and cg01105362, showed direct and significant correlation with fibrosis score (Fig. 1c).

Methylation levels at 19 CpG sites showed a statistically significant association with eGFR (eGFR-DMPs) (Supplementary Fig. 3a; Supplementary Data 2) at an epigenome-wide significance threshold. Of the 19 eGFR-DMPs, 15 had lower methylation levels at high eGFR (Supplementary Fig. 3b). The strongest association was obtained at cg27630540 (on chromosome 4; Bacon corrected $P = 1.8 \times 10^{-9}$), which was in the enhancer region of the UVSSA (UV Stimulated Scaffold Protein A) gene (Supplementary Fig. 3c).

To evaluate the robustness and consistency of the identified methylation changes, we sought to validate the fibrosis-DMPs in two independent external cohorts containing 91[37] and 85 human kidney samples obtained from individuals with and without diabetes and analyzed using Illumina Infinium 450 K arrays. The demographic, clinical, and histopathological characteristics of the two validation datasets can be found in Supplementary Data 3 and 4. Owing to differences in methylation platforms (450 K and EPIC or 850 K arrays), we could directly compare 55 out of our 171 identified fibrosis-DMPs in the first validation cohorts[38]. Overall, the effect estimates of the 55 CpGs in our study strongly and significantly correlated with those described by Gluck et al. (Pearson correlation coefficient $r = 0.81$, $P = 3.8 \times 10^{-14}$; Supplementary Fig. 4a, Supplementary Data 5)[38]. For the second validation cohort (containing 85 kidney samples), Ko et al. identified

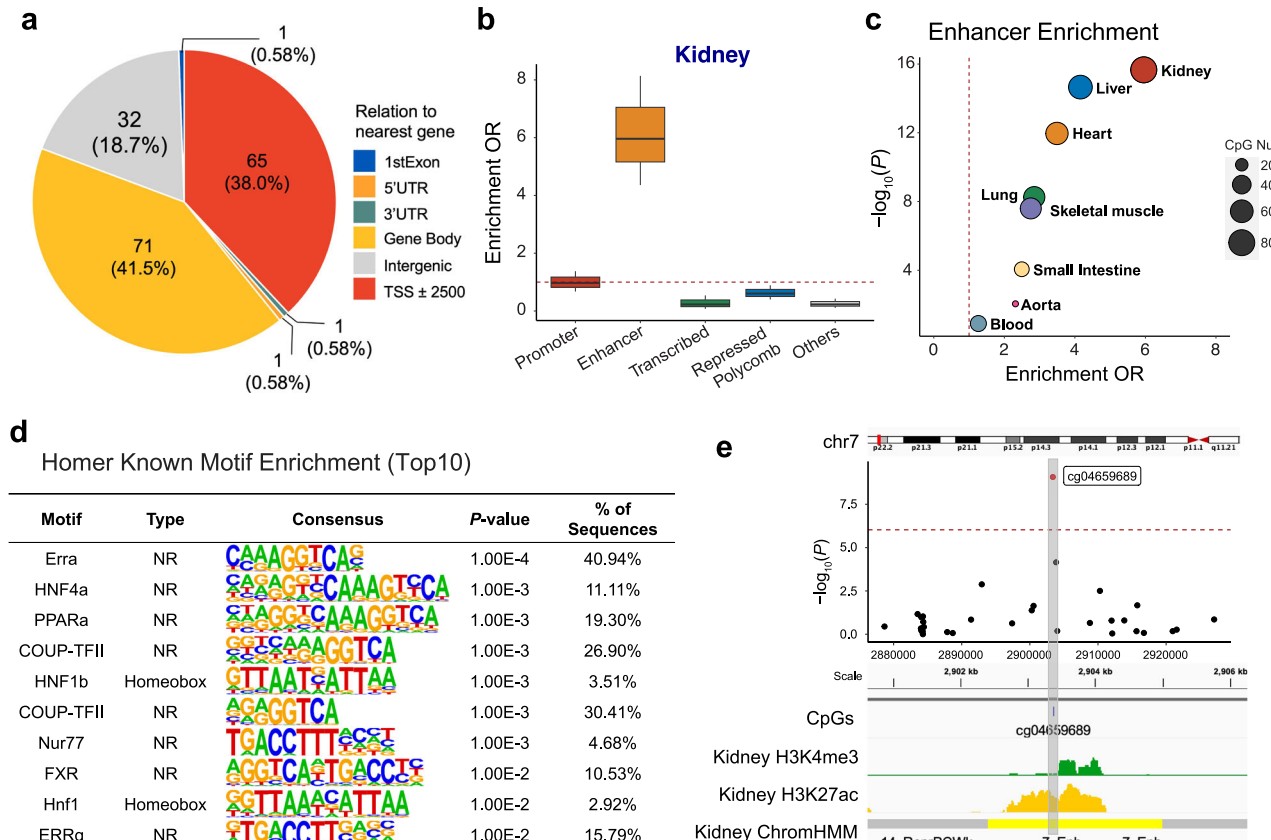

**Fig. 2 | Kidney-disease associated cytosine methylation changes are enriched on kidney enhancers. a** Genomic features of 171 fibrosis-DMPs mapped to Refseq. **b** Functional enrichment of fibrosis-DMPs ($n = 171$) using human kidney chromatin profiling. The fold enrichment of a particular genomic regulatory regions compared to all CpGs present on the EPIC arrays are shown. The x-axis shows genomic annotations and y-axis represents the enrichment odds ratio (two-sided Fisher exact test). Center lines indicate the median fold change, box limits denoted the upper and lower quartiles, and whiskers extend to the 5th and 95th percentiles range. **c** Tissue specificity of fibrosis-DMPs across 8 different human tissue. The functional annotation is based on (Roadmap project) ChromHMM data. The x-axis shows the enrichment OR and the y-axis shows the *P* value calculated by a two-sided Fisher exact test; **d** Transcription factor motif enrichment (based on HOMER) of fibrosis-DMPs. The *P* value was obtained by binomial test. **e** Features of the top significant fibrosis-DMPs cg04659689. Regional association (locusZoom) of the cg04659689 CpG site on chr7p22. The x-axis indicates chromosomal position while the y-axis is the strength of the association ($\log_{10}(P$ value)) derived from EWAS analysis. The cg19942083 probe is shown as a red dot. The lower panel kidney tissue ChromHMM annotation and histone modification marks by chromatin immunoprecipitation (ChIP)-seq. H3K4me3 marks represent active promoters, while H3K27ac marks indicate active enhancer elements.

differential methylation regions using a distinct approach that directly compared methylation ratio between control and diseased tubule samples (binary outcome)[37]. Hence, we rerun the same model using linear regression while adjusting for age, sex, race, diabetes, hypertension, and batch effects to validate the 85 CpG sites out of our 171 identified fibrosis-DMPs which overlap with the second dataset. Similarly, the effect estimates of the 85 CpGs between the two studies were highly consistent (Pearson correlation coefficient $r = 0.73$, $P = 2.0 \times 10^{-15}$), further validating the association between CpG methylation and kidney disease severity (Supplementary Fig. 4b, Supplementary Data 6).

Lastly, we conducted sensitivity analyses to evaluate the robustness of our findings. Sensitivity analyses additionally adjusting for BMI ($n = 370$; Supplementary Fig. 5a, b) or genetic ancestry (the top 5 genetic principal components) ($n = 378$; Supplementary Fig. 5c, d) yielded highly consistent results. Restricting the analysis to participants who had hypertension and/or diabetes ($n = 286$; Supplementary Fig. 5e, f) also yielded consistent associations. Similar results were observed when we limited the sensitivity analysis for those probes showing significant association in the EWAS analysis. In summary, our analysis identified robust changes in cytosine methylation levels in kidney tissue samples associated with eGFR and fibrosis.

## Kidney disease associated cytosine methylation changes are enriched on kidney enhancers

Cytosine methylation changes at transcription factor binding sites could potentially alter the binding strength of transcription factors and decrease gene expression, while methylation in the gene body could increase transcript levels[39]. We used the Reference Sequence (RefSeq) database to annotate transcribed regions, gene body, 5' and 3' untranslated genomic regions to understand the genomic regions showing methylation changes. We observed that the majority of fibrosis DMPs were located in gene body regions (41.5% of DMPs) and regions surrounding the transcription start site (38.0% of DMPs; Fig. 2a).

To further understand the functional role of DMPs associated with fibrosis, we generated adult human kidney-specific functional gene regulatory regions annotation (promoters and enhancers, etc.) by combining multiple histone chromatin immunoprecipitation data (ChiP-seq) using ChromHMM[40]. Investigating the chromatin states where fibrosis-associated DMPs were located, we found that compared to all probes present on the EPIC arrays, fibrosis-DMPs showed the strongest enrichment in regions annotated as enhancers in the human kidney (OR = 5.96, $\chi^2$ test $P < 2.2 \times 10^{-16}$; Fig. 2b using a two-sided Fisher exact test).

Having observed an enrichment of DMPs on enhancers, we analyzed the association between DMPs and enhancers across different tissue types (including liver, heart, and lung) using the Roadmap Epigenomics ChromHMM annotations[41]. We found that kidney fibrosis-associated DMPs were enriched in kidney enhancers ($P < 2.2 \times 10\text{-}16$; Fig. 2c) compared to other tissue enhancers.

Furthermore, to identify transcription factors whose effects may be potentially influenced by cytosine methylation, we performed transcription factor motif analysis using the HOMER software. Compared to all probes on the EPIC array, fibrosis-DMPs were enriched for 15 transcription factor binding motifs (false discovery rate (FDR) < 0.05; Fig. 2d, Supplementary Data 7), with the most prominent motifs representing ERRA, COUP-TFII, NUR77, and FXR, HNF4A, PPARA, HNF1B. Many of these transcription factors have been shown to regulate cellular metabolism, and some have also been proposed to play a crucial role in kidney disease and fibrosis development[42-45]. For instance, we present cg04659689 as an example of one of the top significant fibrosis-DMPs (Bonferroni corrected $P = 8.4 \times 10\text{-}10$) located on a kidney enhancer region (Fig. 2e).

In summary, methylation changes associated with kidney disease were enriched at active functional elements in the kidney. These findings imply the functional importance of the identified fibrosis-DMPs.

## Disease associated genetic variants often drive observed disease associated methylation changes

One of the key limitations of EWAS studies is that phenotype-associated methylation changes are likely the consequence of the disease process rather than the cause. Genetic variants do not suffer from reverse causation, so we evaluated genetic variants associated with methylation changes. We analyzed human kidney cis-methylation quantitative trait loci (cis-meQTL) to identify CpG sites that are under the influence of genotype variation (Fig. 3a). We examined single-nucleotide polymorphisms (SNPs) within a cis window of ±1 Mb of the 171 fibrosis-DMPs. A total of 549,142 SNPs around these CpG sites were tested (constituting 673,614 SNP–CpG pairs), and 3,825 significant cis-meQTLs were identified (FDR < 0.01 level; Supplementary Data 8). Among the 171 fibrosis-DMPs, 67 CpG sites (39.2%) had at least one genome-wide significant cis-meQTL SNP (Fig. 3b). Significant cis-meQTL SNPs were mostly found within 100 kb of target DMPs (Fig. 3c), and those DMPs were more likely to be enriched in kidney enhancer regions (OR = 6.99, $P = 9.0 \times 10^{-14}$), overall suggesting that the observed methylation changes are strongly impacted by genetic variations.

To understand whether the underlying genetic variation-driven methylation changes may be associated with disease development, we overlapped genetic variants that drove methylation differences (3,825 cis-meQTLs) with genetic variants associated with phenotype development (GWAS) (Supplementary Data 9). We identified 43 cis-meQTLs (6 CpG) associated with eGFR GWAS[6], 23 cis-meQTLs (3 CpG) associated with urate GWAS[46], and 7 cis-meQTLs (1 CpG) associated with UACR GWAS[47]. Furthermore, by mapping to additional disease-associated GWAS results, we found 9 cis-meQTLs (6 DMP) associated with blood pressure and 32 cis-meQTLs (3 DMPs) associated with T2DM[48,49].

Finally, to understand the target genes of genotype-associated methylation changes, we examined gene expression changes driven by genetic variations, in expression quantitative trait loci (eQTL) analysis[6]. We found that from the methylation-driving SNPs, 881 unique SNPs showed statistically significant associations with 53 genes, including FBRSL1, OAF, LIMK1, TRIM29, and PXMP2 (Supplementary Data 10). For instance, rs2444239, a cis-meQTL associated with DNA methylation at cg15412087, was also associated with albuminuria (measured as urinary albumin creatinine ratio) and exhibited a significant correlation with OAF expression ($P = 5.04 \times 10^{-18}$; Fig. 3d). Cg15412087 was located in the enhancer region of OAF, as indicated by the H3K27Ac

ChIP-seq signal and chromatin state annotations (Fig. 3e). OAF (Out At First Homolog), a protein-coding gene expressed in the renal cortex, has been reported to be involved in protein reabsorption in the kidney[47]. Knockdown of its orthologs in Drosophila nephrocytes reduces albumin endocytosis[47]. Collectively, our findings indicate that close to a third of kidney disease-associated methylation changes are under genotype influence, and in multiple cases, these underlying genetic variants are also associated with kidney dysfunction. These results suggest that genetic variations play a significant role in driving methylation changes, which in turn can influence disease development.

## Changes in human kidney single cell open chromatin regions in disease states

Epigenetic changes are highly cell-type specific, and although bulk methylation differences have been linked to kidney-specific enhancer changes, the underlying cell types have not been identified. To understand cell-type specific epigenetic variations, we reanalyzed single-nucleus transposase-accessible chromatin with sequencing (snATAC-seq) data from 14 human kidney samples[50]. The snATAC-seq libraries were prepared following the 10x platform and the raw sequencing data were aligned using Cell Ranger ATAC and further analyzed using Signac[51]. After quality control, a total of 401,395 accessible chromatin regions (also known as 'ATAC peaks') were identified in all cell types.

Chromatin accessibility information, which indicates transcriptionally active (i.e., open) and inactive (i.e., condensed) regions, allowed for grouping cells into discrete cell clusters based on differentially accessible chromatin regions (DAR). Using open chromatin-based gene activity indices we were able to identify 24 key kidney cell types, including different types of proximal tubules segments 1-3 (PT_S1, S2, S3, and injured), descending loop of Henle (DLOH), podocytes, cortical and medullary thick ascending loop of Henle (C_TAL and M_TAL), distal convoluted tubule (DCT), connecting tubule (CNT), principal cells of collecting duct (PC), intercalated cells type alpha and beta (IC_A and IC_B), stromal, and different types of immune cells (Fig. 4a; Supplementary Fig. 6a; Supplementary Data 11).

Out of the 14 analyzed samples, 7 can be considered healthy and 7 with CKD based on eGFR and histopathological information (greater than 10% fibrosis). To gain a deeper understanding of cell-type-specific differentially accessible regions (DAR), we compared CKD and control samples, in each cell type. In total, we identified 38,316 cell-specific differentially accessible chromatin regions at a 5% FDR q-value and an absolute log-fold-change threshold of 0.25 (Fig. 4b; Supplementary Data 12). Overall, a higher number of accessible chromatin regions were observed in CKD samples compared to healthy controls. Of the identified differential accessible regions (DARs), the majority ($n = 19,456$; 50.8%) were located in promoter regions (Fig. 4c; Supplementary Fig. 6b), while a minority ($n = 6,264$; 16.4%) were intergenic. Notably, identified DARs between healthy and CKD samples highlighted PT, PC, and collecting duct cell types with the highest numbers of DARs. Proximal tubule cells (PT_S1, PT_S2, PT_S3, and injured_PT) had the greatest number of DARs ($n = 27,566$ in total), followed by the PC ($n = 20,684$), IC_A ($n = 11,154$), and mesangial cells ($n = 5,114$). Amongst the identified DARs, 52.6% ($n = 20,157$) were cell-type specific (only observed in one cell type) while a subset of DARs ($n = 18,159$, 47.4%) were shared between cell types.

Disease-associated cell-specific DARs were enriched for distinct transcription factors (the full list of motif enrichment is provided in Supplementary Data 13), including key cell type-specific transcription factors like HNF4A in the proximal tubules. Transcription factors enriched in disease DARs provide insight into cell-specific signaling pathways altered in CKD. Notably, injured PT DARs were significantly enriched for NF1, HNF1B, and FOS motifs (NF1 $P = 1.0 \times 10^{-266}$; HNF1B $P = 1.0 \times 10^{-214}$; FOS $P = 1.0 \times 10^{-204}$), known regulators of PT identity,

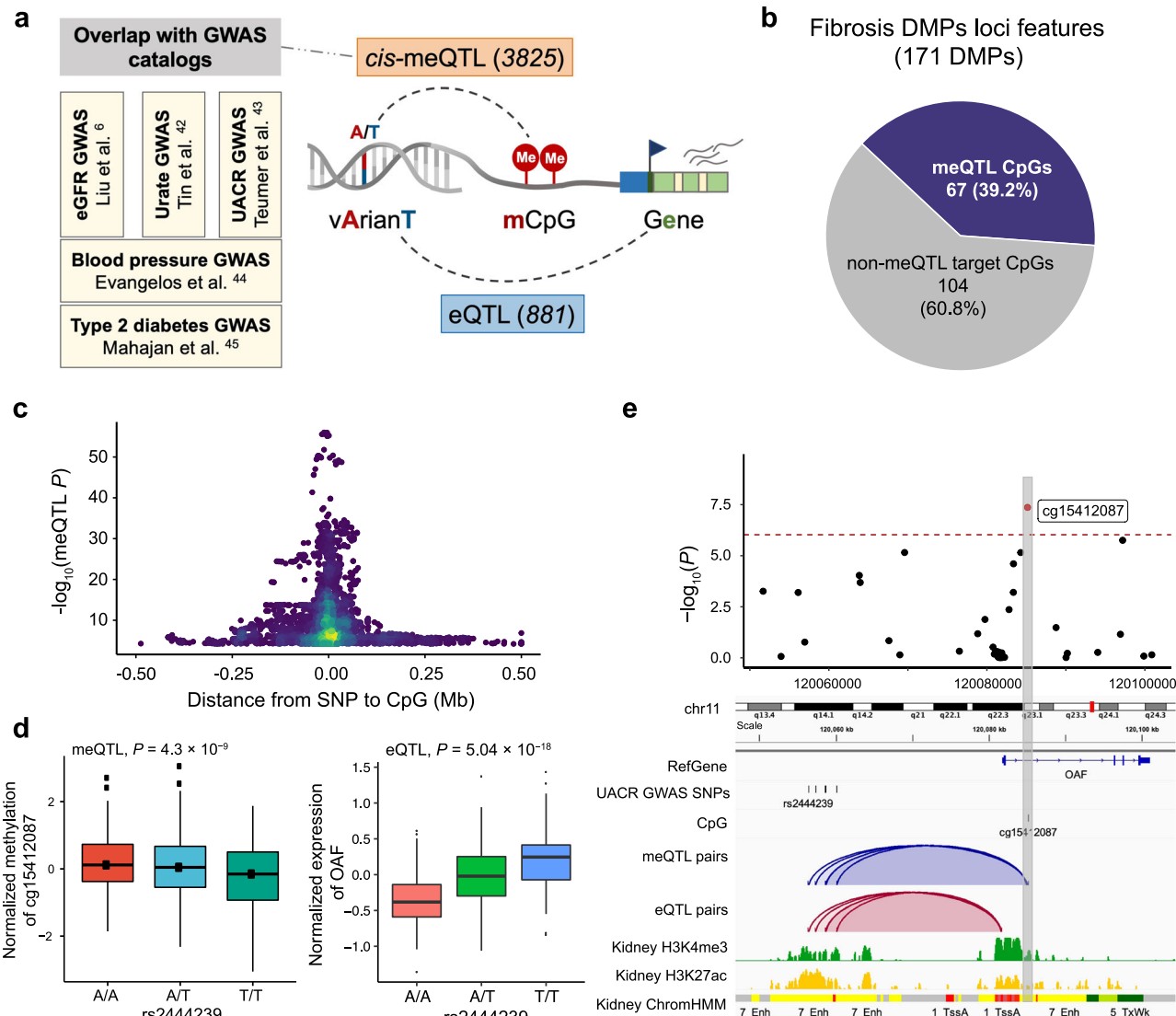

**Fig. 3 | Genetic variation contributes to methylation and corresponding gene expression changes. a** Integrative analysis to identify methylation and disease-associated genetic variants. Parts of the right panel have been created using BioRender.com. **b** Genetic variation likely contributes to methylation changes at 40% of fibrosis-DMPs. **c** Methylation and driving genotype change association. The x-axis is the distance between SNP and their target CpG sites and the y-axis indicates the strength of the meQTL association (-log$_{10}$(*P* value)). **d** The left panel shows the association of genotype (rs2444239, x-axis) and normalized CpG methylation (cg15412087, y-axis) in human kidneys (*n* = 443). The right panel shows the association of genotype (rs2444239, x-axis) and normalized gene expression (OAF, y-axis) in human kidney tubule samples (*n* = 356). Each dot represents a sample,

which was grouped according to the genotype. Center lines indicate the medians, box limits show the upper and lower quartiles, and whiskers extend to the 5th and 95th percentiles. The *P* value (two-sided) was derived from either meQTL regression models or published eQTL meta-analysis of 686 samples. **e** Feature of fibrosis-DMPs cg15412087 and its driving genetic changes (meQTLs). The upper panel shows the regional association of the cg15412087 CpG site on chr11q23. The x-axis indicates chromosomal position while the y-axis is the strength of the association derived from EWAS analysis (-log$_{10}$(*P* value)). The cg15412087 probe is shown as a red dot. The lower panel includes meQTLs, eQTLs, human kidney histone modifications, and chromatin states annotations. The top meQTL (rs2444239) selected based on the *P* value of meQTL associations is highlighted by the label.

suggesting that epigenetic regulation may have implications for the development of kidney disease.

For example, our analysis revealed a subgroup of DARs consisting of 24 regions located within or in close proximity to the PRKAG2 gene (protein kinase AMP-activated non-catalytic subunit gamma 2). Notably, these regions exhibited a more accessible chromatin in CKD. PRKAG2 encodes the γ2 regulatory subunit of the AMP-activated protein kinase (AMPK), a crucial regulator involved in glucose and lipid metabolism, ion and water transport, inflammation, and stress response. Additionally, we observed a significant association between rs10224210, a genetic variant identified in eGFR GWAS[6], and the expression of the PRKAG2 gene. These findings suggest a potential involvement of the identified DARs in modulating kidney function and kidney disease development.

In summary, our human kidney snATAC analysis identified cell-type and disease-specific changes in open chromatin, some of which were associated with genetic variants known to play a role in disease development, indicating the potential causal role for open chromatin and methylation changes.

### The relationship between open chromatin and methylation changes

Next, we wanted to understand the relationship between methylation- (DMPs) and single-cell open chromatin- changes. We found that 145 of the 171 fibrosis-associated DMPs were in open chromatin regions in human kidney single cell data. Fibrosis-DMPs showed enrichment in open chromatin regions in PT, thin descending loop of Henle (DLOH), and principal cells (PC), with the highest enrichment in PT_S3

**a**   Human kidney snATAC-seq
        *n* = 80,845 cells

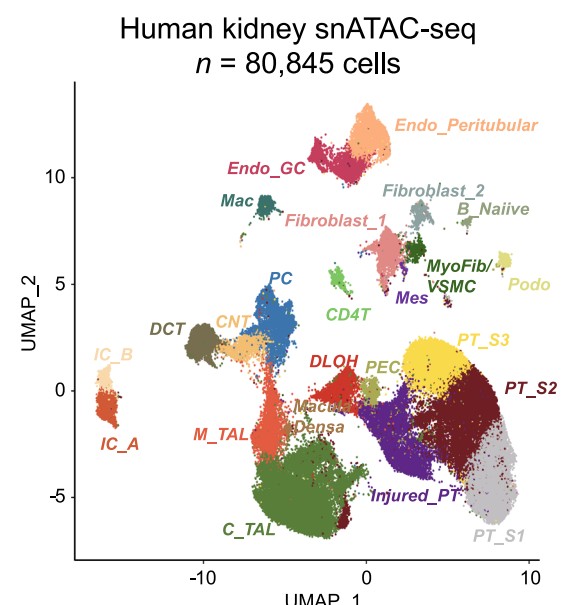

**b**   Number of DARs in each cell type

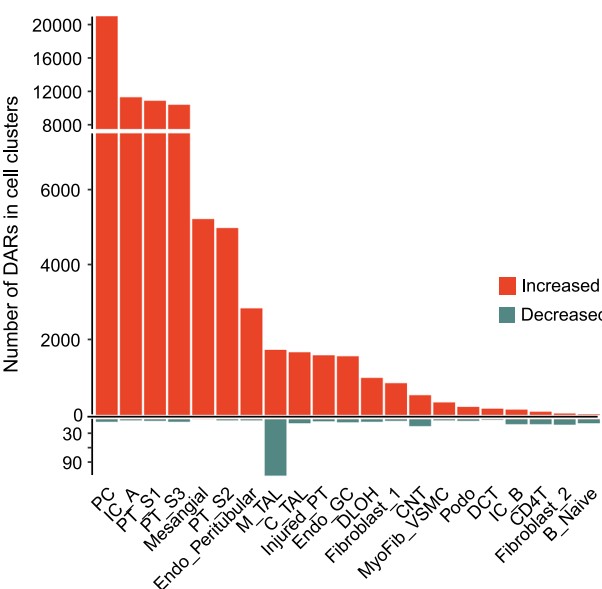

**c**   Genome features of DARs

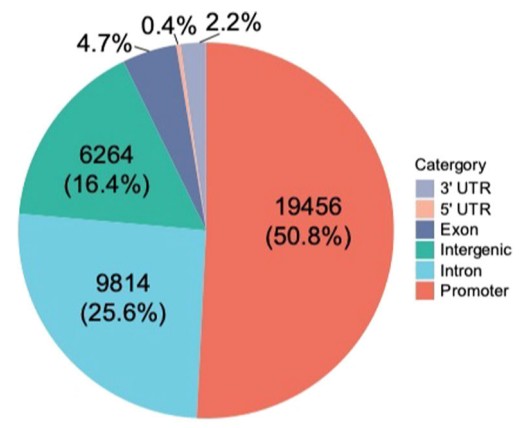

**d**   Enrichment of DMPs in open chromatin regions

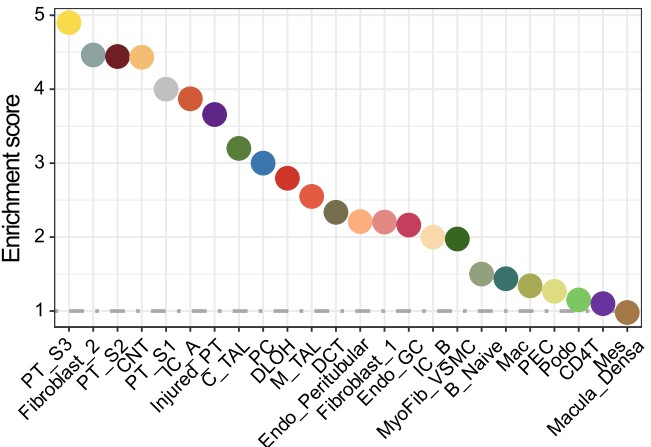

**Fig. 4 | DMPs are enriched at open chromatin regions and CKD differential accessible regions in kidney proximal tubules. a** Single-nucleus accessibility UMAP for 80,845 human kidney cells by snATAC-seq. Each dot represents a cell, with color coding for the cell types: Endo_GC; endothelial cells of glomerular capillary tuft, Endo_peritubular; endothelial cells of peritubular vessels, Mes; meseangial cells, VSMC/Myofib; vascular smooth muscle cells/myofibroblast, PEC; parietal epithelial cells, Podo; podocyte, PT_S1; proximal tubule segment 1, PT_S2; proximal tubule segment 2, PT_S3; proximal tubule segment 3, Injured_PT; injured proximal tubule cells, DLOH; thin descending loop of Henle, C_TAL; cortical thick ascending loop of Henle, M_TAL; medullary thick ascending loop of Henle, DCT; distal convoluted tubule, CNT; connecting tubule cells, PC; principal cells of collecting duct, IC_A; Type alpha intercalated cells, IC_B; Type beta intercalated cells,

CD4T; T lymphocytes CD4 +, B_Naiive; Naiive B lymphocyte, Mac; macrophage. **b** Number of differentially accessible regions (DARs) in each cell type in disease state. The x-axis represents the cell types, while the y-axis indicates the count of significant DARs. The color scale indicates the direction of chromatin accessibility changes compared to control samples, with red indicating increased accessibility and blue indicating decreased accessibility. **c** The distribution of disease-associated DARs across the RefSeq genome. **d** Enrichment of fibrosis-DMPs located in cell-types open chromatin regions compared to all CpGs present on the EPIC arrays. The x-axis shows the cell types and y-axis represents the enrichment OR, which was examined using a two-sided Fisher exact test. The color of the dot indicates the specific cell type.

(OR = 4.90, $P < 2.2 \times 10^{-16}$; Fig. 4d) compared to probes present on the EPIC array. Genomic regions harboring fibrosis-associated DMPs in PT cells were enriched for ERRA, PPARa, RXR, as well as HNF4a transcription factor motifs (the full list of motif enrichment is provided in Supplementary Data 14) again indicating their potential functional importance.

We further overlapped the 171 fibrosis-associated DMPs with cell-specific DARs, and found that approximately one-fifth of DMPs were located in open chromatin regions that changed during disease state; disease-associated DAR regions ($n = 53$, 31.0%). Most of these regions

exhibited a more accessible chromatin status in CKD. Notably, the overlap highlighted PT, PC, and collecting duct cells harboring the highest numbers of DMPs. These regions were enriched for transcription factors motifs including JUND, KLF5, FRA1, JUNB, and ETS: RUNX (the full list of motif enrichment is provided in Supplementary Data 15). These transcription factors were known to regulate cell proliferation, apoptosis, tubulointerstitial inflammation, and fibrosis. These findings further support the potential role of epigenetic alterations in transcriptional regulation and kidney disease development.

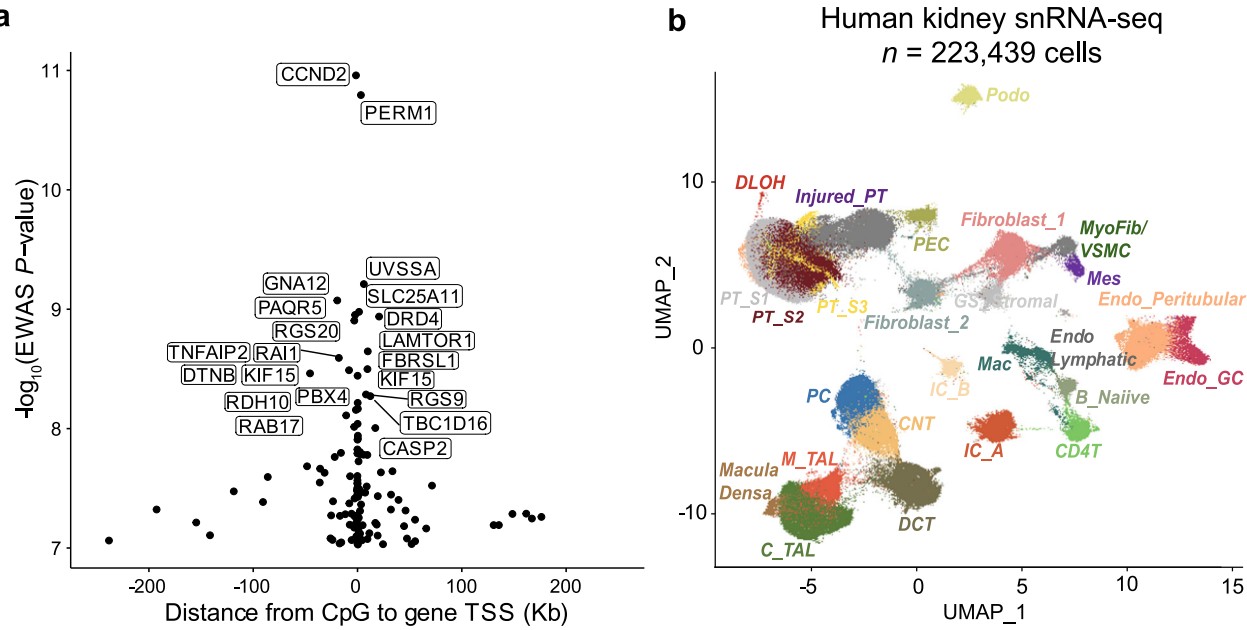

**a**

**b** Human kidney snRNA-seq
*n* = 223,439 cells

**c**

125 fibrosis-DMP target genes showing cell-type expression

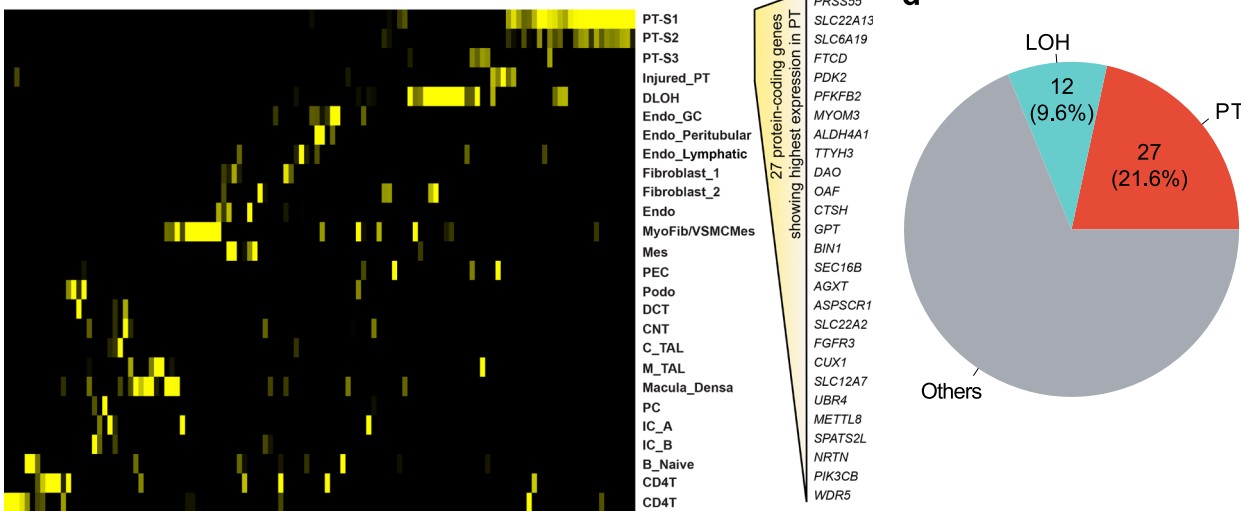

**d**

**Fig. 5 | Fibrosis-DMPs nearest gene expressed in kidney proximal tubules. a** The location of the prioritized gene promoter in relation to CpG sites. The x-axis is the distance from CpG sites to their target gene TSS and the y-axis is the strength of the association derived from EWAS analysis (-log$_{10}$(*P* value)). The prioritized genes of top 20 significant fibrosis-associated DMPs are labeled. **b** Single-nuclear RNA-seq UMAP of 223,438 human kidney cells. Each dot represents a cell, with color coding for the cell types: Endo_GC; endothelial cells of glomerular capillary tuft, Endo_peritubular; endothelial cells of peritubular vessels, Endo_Lymphatic; endothelial cells of lymphatic vessels, Mes; meseangial cells, VSMC/Myofib; vascular smooth muscle cells/myofibroblast, PEC; parietal epithelial cells, Podo; podocyte, PT_S1; proximal tubule segment 1, PT_S2; proximal tubule segment 2, PT_S3; proximal tubule segment 3, Injured_PT; injured proximal tubule cells, DLOH; thin descending loop of Henle, C_TAL; cortical thick ascending loop of Henle, M_TAL; medullary thick ascending loop of Henle, DCT; distal convoluted tubule, CNT; connecting tubule cells, PC; principal cells of collecting duct, IC_A; Type alpha intercalated cells, IC_B; Type beta intercalated cells, CD4T; T lymphocytes CD4 +, B_Naiive; Naiive B lymphocyte, Mac; macrophage, GS_Stromal; cells specifically present in sclerosed glomeruli. **c** Heatmap of the expression of 125 co-access genes in each cell type (yellow: high expression, black: low expression); **d** The number of top 2 cell types expressing the highest fibrosis DMP target genes.

## The relationship between epigenetic and gene expression changes

Next, we wanted to identify potential target genes for the observed methylation changes. First, we annotated DMPs with the nearest protein-coding gene. Among the 171 fibrosis DMPs, 137 DMPs were annotated to 125 protein-coding genes, including CCND2, SLC25A11, FBRSL1, and KIF15 (Fig. 5a; the full list is provided in Supplementary Data 16). Notably, one of the most significant fibrosis-associated DMP; cg20691282, was annotated to the PERM1 gene (PPARGC1 and ESRR

induced regulator, muscle 1 gene), which has been demonstrated to enhance mitochondrial biogenesis and metabolism[52].

We next aimed to interrogate cell types in which methylation target genes expressed. We reanalyzed gene expression information from 17 adult human kidney samples including 10 control samples and 7 with CKD[50]. The single-nucleus RNA sequencing (snRNA-seq) libraries were processed using Cellranger (10X Genomics) and analyzed with Seurat. Prior to analysis, doublets were removed using DoubletFinder, and batch effects were corrected using Harmony[53–55]. A total of 223,439

cells passed quality control filters, capturing all major kidney cell types in both healthy and diseased conditions across different anatomical regions (Fig. 5b; Supplementary Fig. 6c).

By analyzing the expression patterns of the 125 targeted genes, we found that the majority of genes exhibited enrichment in the kidney cortex, suggesting that their alterations predominantly affect the function and physiology of this specific region (Fig. 5c). Furthermore, our analysis revealed distinct expression patterns of these genes across different cell types within the kidney, indicating that their dysregulation may be cell type-specific. Interestingly, methylation differences were more common around genes expressed in proximal tubule (PT) cells. Moreover, we found 27 genes (21.6%; Fig. 5d) exhibiting their highest expression in PT cells (PT_S1, PT_S2, PT_S3, and Injured_PT), including those associated with metabolism (e.g., SLC22A13, SLC6A19, FTCD, ALDH4A1, and OAT) and fibrogenesis (e.g., PDK2, PFKFB2, MYOM3). These findings suggest that kidney structure-associated DNA methylation can impact gene expression in various cell types, particularly in PT cells.

Taken together, our results provide valuable insights into the cell-type specificity of DNA methylation changes in kidney disease. They underscore the importance of considering the unique cellular contexts in which these alterations occur, shedding light on the potential mechanisms driving kidney pathophysiology.

## Fibrosis-correlating DMPs associated with the expression of solute carriers and metabolic genes

To understand the functional role of genes showing methylation and open chromatin changes in disease, we analyzed the function of the targeted genes. Gene ontology analysis of epigenetically targeted genes revealed 64 biological processes ($P < 0.05$), such as gluconeogenesis, response to insulin stimulus, and regulation of the MAPK cascade (Fig. 6a; Supplementary Data 17). Impaired renal gluconeogenesis is a key feature of DKD. Reactome pathway enrichment analysis identified 41 pathways ($P < 0.05$; Supplementary Data 18), including metabolism (such as amino acids and derivatives, glyoxylate, and glucose), transporter-related signaling pathways (such as Na + /Cl-dependent neurotransmitter transporters, bile salts and organic acids, metal ions and amine compounds, and inorganic cations/anions and amino acids/oligopeptides), and gluconeogenesis, as well as MAPK1/MAPK3 signaling. Some of these processes are known to play a role in kidney fibrosis.

Due to the limited number of the targeted genes in this dataset, we expanded gene ontology analysis for the top 1,000 fibrosis-DMPs (829 genes). Gene ontology analysis indicated enrichment for glucose metabolism, potassium ion transportation, as well as regulation of MAPK signaling (Supplementary Fig. 7a). Subsequent pathway enrichment identified 29 pathways (FDR < 0.05; Supplementary Fig. S7b), including amino acid metabolism (such as histidine, tryptophan, and arginine), protein digestion and absorption, gluconeogenesis, cell adhesion, and other metabolic pathways. In summary, we observed that differentially methylated sites in disease showed enrichment for open chromatin regions in kidney proximal tubules and identified multiple genes, mostly solute carriers those methylation changes were not only associated with kidney disease but also with the expression of a variety of transporters.

Here we highlighted the cg09837037 site, which showed strong associations with fibrosis in EWAS analysis. The cg09837037 site was located at the SLC6A19 promoter, which was at an open chromatin region in PT cells (Fig. 6b). Additionally, we observed negative correlations between the methylation level of cg09837037 and SLC6A19 expression (Pearson correlation coefficient r = −0.56, $P < 2.2 \times 10^{-16}$; Fig. 6c). Furthermore, SLC6A19 gene expression in human kidneys significantly correlated with fibrosis (r = −0.53, $P < 2.2 \times 10^{-16}$). SLC6A19 encodes a sodium-coupled transporter located at the apical membrane of renal proximal tubule cells and is prominently involved in the absorption of neutral amino acids. Mutations in SLC6A19 have been associated with aminoaciduria. Another similar example is the cg06224737 site, located 365 kb from the TSS of the SLC22A13 gene which has been implicated in phosphate and urate reabsorption in human kidney tubules[56,57]. The methylation level of cg06224737 was found to be negatively correlated with the expression of SLC22A13 (Pearson correlation coefficient r = −0.61, $P < 2.2 \times 10^{-16}$; Fig. 6d). SLC22A13 gene expression in human kidneys was inversely correlated with fibrosis (Pearson correlation coefficient r = −0.51, $P < 2.2 \times 10^{-16}$). In summary, we observed that differentially methylated sites in disease showed enrichment for open chromatin regions in kidney proximal tubules and identified multiple genes, mostly solute carriers, whose methylation changes were not only associated with kidney disease but also with the expression of various transporters.

## Methylation risk scores improve kidney function and prognosis estimation

Our analysis indicated that methylation changes offer an important combined read-out for both genetic and non-genetic factors (environmental) and demonstrate an association with disease state. In recent years, genetic factors integrated as polygenic risk scores (PRS) have emerged as powerful tools for predicting diseases. We hypothesized that methylation risk scores (MRS) might work even better than PRS since they integrate both genetic and non-genetic factors. As many of the fibrosis-DMPs were correlated, we applied LASSO regression to narrow down the number of disease-associated CpGs and computed weighted MRS for everyone using a linear combination of the effect estimates and methylation levels of these selected CpGs (Fig. 7a).

First, we wanted to understand whether MRS can improve fibrosis estimates in the kidney. As noted before, eGFR poorly correlates with fibrosis (r of −0.46) and it is hard to estimate the degree of fibrosis based on clinical parameters. We computed a weighted MRS based on 19 LASSO-selected CpGs (Supplementary Data 19) from 70% of the study samples (the training set) for individuals. We used the remaining 30% of the samples as study samples (the testing set). We then generated log2-transformed fibrosis estimates in the testing set using a polynomial linear regression model, with eGFR as the predictor variable, controlling for age, sex, race, diabetes mellitus, and hypertension status. This eGFR-based prediction model explained 30.3% of the log2-transformed fibrosis level variance, consistent with previous studies, indicating that clinical parameters relatively poorly predict histological damage. The addition of MRS into the base model significantly improved this capability to 42.1% (likelihood ratio χ2 test $P = 6.3 \times 10^{-6}$; Fig. 7b), indicating that the established MRS can enhance the accuracy of fibrosis estimation.

Next, we wanted to know whether MRS can help to identify CKD cases. We fine-tuned the MRS weights using all available 399 samples. We conducted a logistic regression analysis, controlling for age, sex, race, diabetes mellitus, and hypertension status. MRS was significantly associated with CKD incidence (OR = 2.57, 95% CI = 1.37- 4.95; $P = 3.6 \times 10^{-3}$; Fig. 7c). Compared to the initial model that only included age, sex, race, presence of DM, and/or HTN, the addition of MRS significantly improved the discriminative performance of incident CKD (area under the curve, 0.76 versus 0.70, likelihood ratio χ2 test $P = 1.3 \times 10^{-7}$). To assess the robustness of our results, we conducted sensitivity analyses for the associations between MRS and CKD using the 2021 (race-free) eGFR equation[58]. We identified 110 CKD patients using the 2021 (race-free) eGFR equation. Remarkably, the findings remained consistent. The established MRS was significantly associated with new CKD incidence (OR = 2.61, 95% CI = 1.37 − 5.05; $P = 3.7 \times 10^{-3}$). Compared to the initial model that only included age, sex, race, presence of DM, and/or HTN, the addition of MRS significantly improved the discriminative performance of incident CKD (area under the curve, 0.74 versus 0.70, likelihood ratio χ2 test $P = 1.4 \times 10^{-5}$) reaffirming the robustness of our results.

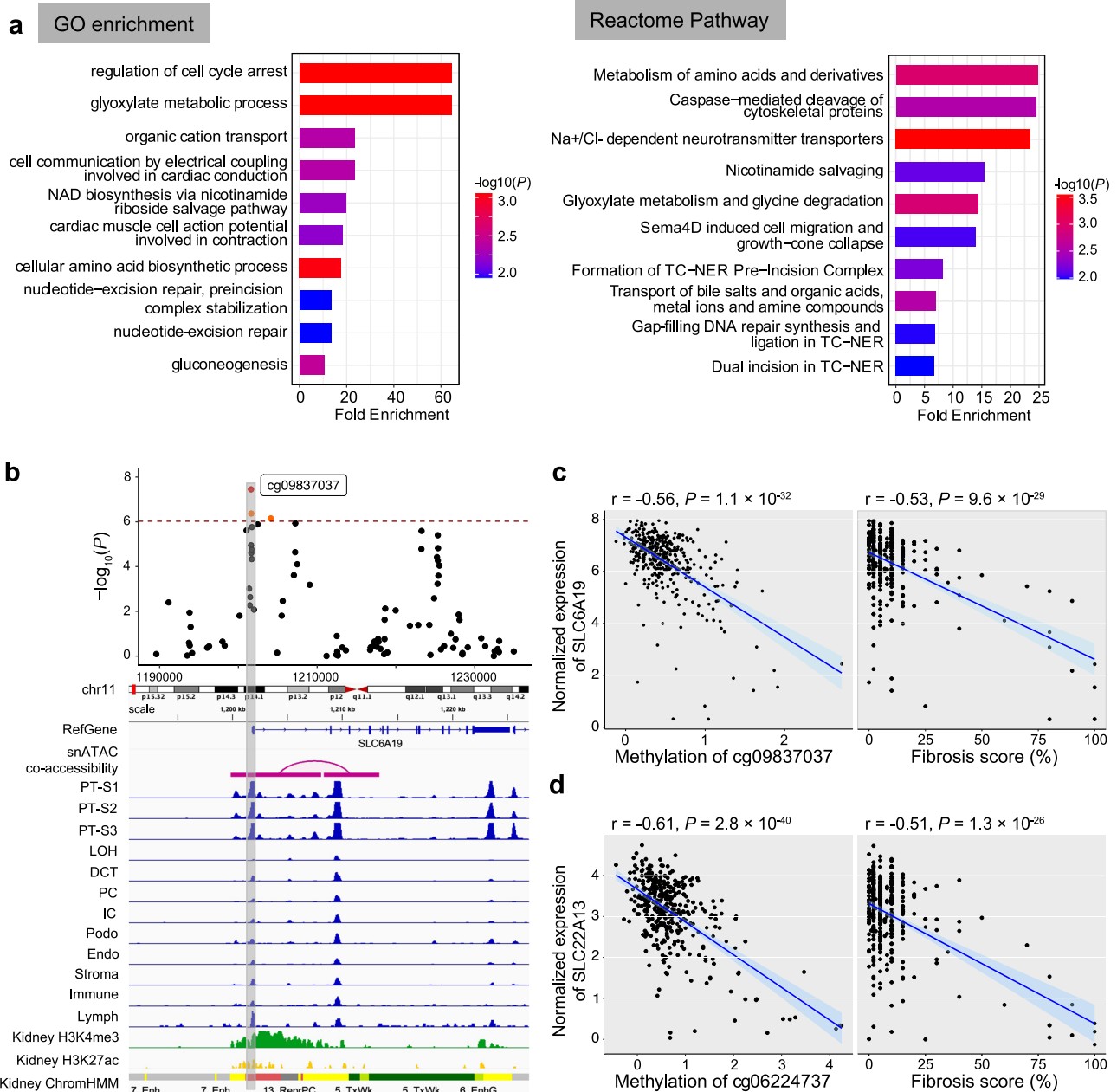

**Fig. 6 | DMPs are correlated with the expression of solute carriers and metabolic genes. a** Functional enrichment of fibrosis-associated DMPs' nearest genes. The left panel shows Gene Ontology (GO) enrichment, while the right panel depicts the Reactome Pathway enrichment. The top 10 terms passing a nominal *P* value < 0.05 are shown. The y-axis shows the enriched GO term or Reactome pathways ordered by enrichment scores (x-axis), and the color indicates the strength of enrichment (-log10(*P* value)) from strongest (red) to lowest (blue). **b** Feature of fibrosis-DMPs cg09837037 and its prioritized target gene SLC6A19. The upper panel shows the regional association of the cg09837037 CpG site on chr11p15. The x-axis indicates chromosomal position while the y-axis is the strength of the association derived from EWAS analysis ($-\log_{10}(P$ value)). The cg09837037 probe is shown as a red dot and highlighted in a box to facilitate mapping across tracks. The lower panel includes Cicero connections, human kidney snATAC-seq

chromatin accessibility, histone modifications, and chromatin states. **c** Left panel: correlation between methylation levels at cg09837037 (x-axis) and SLC6A19 gene expression (y-axis). Right panel: correlation between fibrosis score (x-axis) and SLC6A19 gene expression (y-axis). Each data dot represents one kidney sample. The correlation coefficient and two-sided *P* value were obtained from the Pearson correlation (r) statistic test. The shaded area indicates the 95% confidence interval for the correlations. **d** Left panel: correlation between methylation levels at cg06224737 (x-axis) and SLC22A13 gene expression (y-axis). Right panel: correlation between fibrosis score (x-axis) and SLC22A13 gene expression (y-axis). Each data dot represents one kidney sample. The correlation coefficient and two-sided *P* value were obtained from the Pearson correlation (r) statistic test. The shaded area indicates the 95% confidence interval for the correlations.

Lastly, we examined whether MRS could predict future kidney function decline in our cohort. Our cohort included 117 participants with at least 3 months of follow-up data after kidney sample collection. A mixed-effect model was used to identify factors associated with longitudinal eGFR levels, controlling for known clinical covariates and

incorporating random-effects terms to account for differences in eGFR values between subjects and over time. The results showed that fibrosis MRS was significantly associated with longitudinal eGFR changes (beta = −18.7, $P = 6.7 \times 10^{-7}$; Fig. 7d). Furthermore, compared to the model where only clinical parameters were included, the

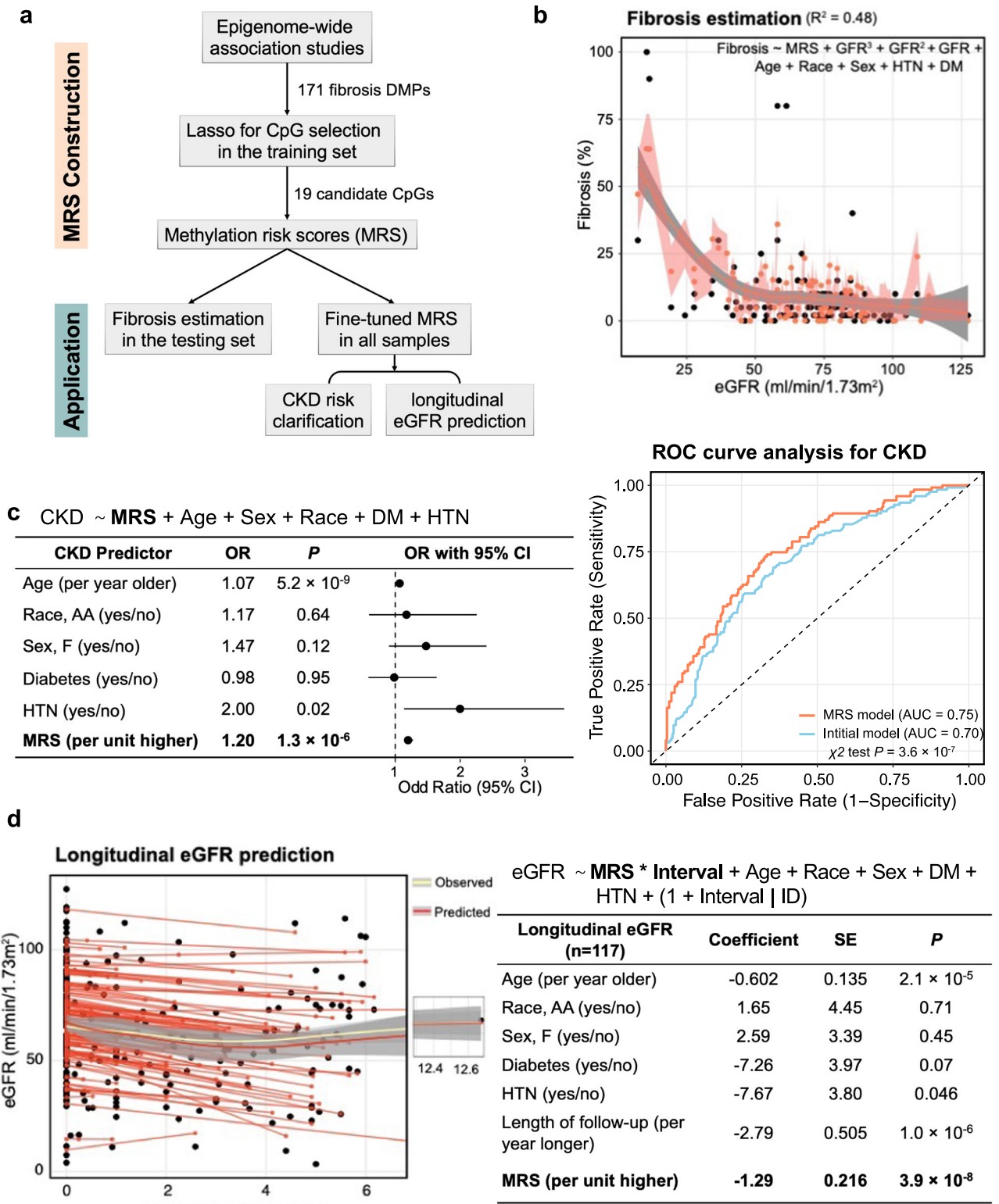

inclusion of MRS significantly improved the model's fitness (likelihood ratio test $P = 6.1 \times 10^{-11}$, Akaike Information Criterion 2062.4 versus 2111.9, $R^2$ 0.77 versus 0.67).

These findings suggest that fibrosis-related methylation information is clinically important for kidney function estimation and long-term prediction, highlighting the potential utility of MRS as a prognostic tool for CKD.

**Methylation differences predict longitudinal eGFR change**

We further wanted to elucidate whether methylation alterations could predict kidney disease development by analyzing a subset of 117 participants with longitudinal eGFR data available. To account for variability in the follow-up time and baseline eGFR, we used a linear mixed model that includes an individual-specific random effect for our analysis.

**Fig. 7 | Methylation risk scores improve Kidney Function and prognosis estimations. a** Schematic representation of the construction and application of the methylation risk score (MRS). The MRS CpG was selected by LASSO models based on the 171 fibrosis DMPs. **b** MRS-based model estimated fibrosis score. The weighted MRS was computed based on 19 LASSO-selected CpGs from the training set for each individual in the testing set. The model explains 40.2% variance of log2-transformed fibrosis. The black dots represent the observed value and the coral dot indicates MRS-model predict value. The shaded area indicates the 95% confidence interval for the predicted values. **c** Left panel: MRS independently predicts the incidence of CKD. Right panel: Discriminative performance of MRS predicting

incident CKD based on the model only included age, race, gender, DM, and HTN status. Adding MRS provided additional information and improved the discrimination power (area under the curve [AUC] = 0.76 vs 0.70, $P = 8.8 \times 10^{-8}$). **d** Left panel: Performance of MRS predicting future kidney function based on the model included age, race, gender, DM, and HTN status ($R^2$ of 0.77). The x-axis represents the follow-up time in years, while the y-axis displays the eGFR values. The yellow curve illustrates the observed overall eGFR change, and the red curve depicts the predicted overall eGFR change pattern. The shaded area indicates the 95% confidence interval for the fitted curve. Right panel: MRS independently predicts future kidney function decline in the follow-up cohort.

We discovered that the cytosine methylation levels of 15 CpG sites significantly correlated with changes in longitudinal eGFR at epigenome-wide significance ($P < 9.42 \times 10^{-8}$; Supplementary Fig. 8a-b; Supplementary Data 20). The strongest association was observed on chromosome 8 with cg09514524 ($P = 7.56 \times 10^{-9}$), located in the promoter region of glutamic pyruvate transaminase (GPT; Supplementary Fig. 8c), also known as cytosolic alanine aminotransferase 1 (ALT1). ALT1 has been linked to non-alcoholic fatty liver disease and insulin resistance[59].

Of the 15 identified CpG sites, 5 were also classified as fibrosis-DMPs (cg27630540, cg04659689, cg09514524, cg21366655, and cg25544164), and 4 were also eGFR-DMPs (cg27630540, cg04659689, cg01119571, and cg25544164). These findings suggest that CpG methylation may play role in long-term changes in eGFR, thereby providing a new perspective on how epigenetic changes can contribute to kidney disease progression.

In summary, our findings demonstrate that methylation differences at specific CpG sites can predict longitudinal eGFR changes, suggesting a potential role for epigenetic modifications in kidney disease development and progression.

## Discussion

In this study, we conducted the most extensive epigenome-wide analysis of 399 human kidney tissue samples, including controls, diabetic, hypertensive, and CKD tissues. We identified methylation changes associated with disease states. We found that nearly 30% of these changes were influenced by underlying genetic variations, mostly in kidney enhancer regions, associating with (and likely driving) the expression of nearby genes, indicating that these methylation changes likely mediate disease development. We also reanalyzed human kidney single-cell open chromatin changes in healthy and diseased kidneys, identifying a large number of open chromatin changes in disease and small subset of consistent epigenetic (methylation and open chromatin) changes in PT cells associated with metabolism-related gene expression. Finally, we implemented methylation risk scores. MRS were shown to improve disease state annotation and prediction of kidney disease development. Furthermore, methylation differences predicted future kidney function decline.

Epigenetic changes (EWAS) in kidneys of patients can provide critical insight into understanding disease pathogenesis. Here we provide the largest sample collection to date combined with stringent statistical analysis. Our EWAS analysis identified DMPs associated with kidney fibrosis, with many of these methylation changes enriched in kidney-specific regulatory regions, such as enhancers. The use of human kidney single-cell open chromatin information allowed us to prioritize target genes that could be regulated by the identified DMPs. Notably, the majority of the prioritized genes were found to be enriched in proximal tubule cells, supporting the notion that DNA methylation alterations may preferentially impact gene expression in these cells. Future studies shall analyze other cell types as PT cells are the most abundant cell types in microdissected tubule samples. In addition, as methylation patterns are highly cell type-specific, accurate inclusion of cell fraction will be important. Additionally, our findings suggested that many DMPs were associated with changes in the

expression of genes involved in kidney metabolism, which is known to play a significant role in disease development. It is important to note that one of our objectives was to meticulously dissect the influence of genetic factors from non-genetic factors (often attributed to environmental influences). Through our rigorous analysis, we have arrived at a significant and noteworthy finding: the majority of observed changes in methylation cannot be solely attributed to genetic differences among individuals. While genetic variations undoubtedly play a role in shaping epigenetic patterns, our results demonstrate that these genetic differences alone do not account for the entirety of the observed alterations, suggesting that a multitude of factors beyond genetics must be considered to gain a more holistic perspective on disease pathogenesis.

To complement bulk methylation analysis, we performed epigenome analysis of human disease kidney samples at single-cell resolution by reanalyzing open chromatin regions of over 60,000 cells, defining disease-specific regulatory landscape at single-cell resolution. We observed that the fibrosis DMPs were enriched in open chromatin regions that showed differential accessibility between healthy and CKD samples. Moreover, transcription factor binding enriched in the DMP-located DARs were found to be involved in renal inflammation and fibrosis, further supporting the potential role of epigenetic alterations in kidney disease pathogenesis. In summary, we provide the most comprehensive description of epigenetic changes in human kidney samples in disease state at single-cell resolution. Moreover, we identified a substantial number of open chromatin changes in disease samples, along with a small subset of consistent epigenetic changes (in both methylation and open chromatin) in proximal tubule (PT) cells that were associated with gene expression related to metabolism. It's worth noting that our study primarily focused on analyzing micro-dissected samples enriched for proximal tubules. As a result, further investigations using whole kidney tissue are warranted to provide additional insights into cell-specific changes.

Lastly, and most importantly we show the promise of the development of methylation risk scores. As methylation changes act as integrators of both genetic and environmental changes methylation risk scores have the potential improve our precision of diagnosing and predicting kidney disease. Our methylation risk scores can precisely estimate disease state and severity in the kidney. Our analysis of longitudinal eGFR data revealed that the methylation levels of certain CpG sites were significantly correlated with changes in eGFR over time. Some of these CpG sites were also classified as fibrosis-DMPs or eGFR-DMPs, providing evidence that DNA methylation changes may have a role in long-term eGFR changes and potentially contribute to kidney disease progression. It's worth noting our study's reliance on kidney biopsy data is integral to the development and application of the MRS. While the MRS holds great promise in enhancing the precision of kidney disease diagnosis and prediction, it is important to recognize that obtaining kidney biopsy samples may not always be feasible or suitable for all individuals. Therefore, future research efforts in non-invasive (such as blood) alternatives for obtaining the data are needed to expand the accessibility and utility of the MRS in clinical practice.

Our study is also subject to several limitations that merit acknowledgment. First, we were limited by the availability of albumin-

to-creatinine ratio (ACR) data, which was only available for fewer than one-third of the participants. As a result, we did not include ACR in the CKD incidence and longitudinal eGFR decline prediction models. Second, we conducted an overlap analysis between genetic variants influencing methylation differences and genetic variants previously associated with the development of specific phenotypes, however, we recognize that causative relationships and the precise mechanisms underlying these associations warrant further investigation. Third, we were constrained by the availability of electronic health record data from an independent cohort. Consequently, we had to optimize the utilization of the available dataset, using a 70-30 training-testing split to evaluate the MRS performance on new samples.

While we have provided a comprehensive analysis of human kidney tissue samples, further research is necessary for a longitudinal analysis and understanding the contribution of genotype-driven changes. Future studies should include experimental validation and functional studies to confirm these associations and elucidate the precise mechanisms by which DNA methylation and open chromatin changes contribute to kidney disease. Once developed, single-cell methylation studies will be essential to further enhance our understanding of the role of methylation changes in this context.

In conclusion, our study sheds light on the epigenetic changes in human kidneys, including DNA methylation and single-cell open chromatin changes, genetic variants, gene expression, and kidney disease development. We identified DMPs and cell type-specific open chromatin changes associated with kidney fibrosis and changes in eGFR. We also demonstrated the clinical utility and predictive value of MRS and methylation changes for improving outcome estimations. These findings may ultimately contribute to the development of novel therapeutic strategies and improve our understanding of the molecular pathways involved in kidney disease progression.

## Methods

### Study populations

The primary cohort consisted of a cross-sectional evaluation of 506 human participants undergoing clinically indicated nephrectomies for renal neoplasia. Kidney samples were obtained from a non-neoplastic portion surrounded by at least 2 cm of normal tissue margins via the Cooperative Human Tissue Network. Human kidney tissue collection was approved by the University of Pennsylvania Institutional Review Board, and no informed consent was obtained because the study was deemed IRB-exempt (exemption IV). Demographic, clinical information, and laboratory data were collected through an honest broker. Details of data collection have been previously described[60]. The degree of tubulointerstitial fibrosis was scored (fibrosis score) in an unbiased manner by a specialized renal pathologist using periodic acid–Schiff-stained slides. eGFR values were calculated using the CKD Epidemiology Collaboration equation based on serum creatinine, age, sex, and self-reported or clinician-determined race, as obtained from medical charts[61]. Finally, our epigenome-wide association analyses with fibrosis and eGFR utilized 399 subjects with methylation measurements, kidney histological scores, and good-quality genotype data. Additionally, for a subset of samples (117 subjects), we were able to obtain longitudinal kidney function measurements with at least 3 months of follow-up after nephrectomy.

### DNA methylation profiling and EWAS analysis

Kidney DNA methylation at >850,000 CpG sites was profiled by using Infinium Methylation EPIC BeadChip. A detailed data processing workflow has been previously described[6]. In brief, data pre-processing and quality control were performed using SeSAMe (v1.5.3)[62]. Probes with missing values in more than 20% of samples, non-unique 30 bp 3'-subsequence, low mapping quality, inconsistent extension base fluorophore, extension base SNPs causing a color channel switch, non-CpG sites, and SNPs on chromosomes X, Y, and M, as well as global monomorphic variants over 1%, were excluded. Sample outliers, defined as those exceeding ±3 standard deviations from the mean of the PC1 and PC2 principal components based on PCA of the top 10,000 variant probes, were also excluded. Finally, a total of 701,519 CpG sites from 399 subjects were included for further analysis.

DNA methylation levels were determined using methylation M-values, which are defined as the log2 ratio of the intensities of methylated probe versus unmethylated probe[63]. To account for potential batch effects that may influence DNA methylation measurements, we employed linear mixed-effects models in our analyses. Specifically, we regressed M-values on technical factors including bisulfite conversion control, mean intensity of measurements, sample plate, BeadChip sentrix, and lymphocytic infiltration, and obtained the residuals. For the EWAS analysis, we used log2-transformed kidney fibrosis scores. Subsequently, we used linear regression to assess the association between normalized fibrosis scores and residualized methylation while controlling for potential confounders such as age, sex, race, hypertension status, and diabetes status. To reduce bias and inflation of the EWAS results, we applied the BACON method[64]. After correction, the estimated inflation factors were 0.99 and 1.10 for fibrosis and eGFR, respectively. CKD traits-DMPs were identified at the epigenome-wide significance level ($P < 9.42 \times 10^{-8}$)[65].

To test the robustness of our findings, we conducted three separate sensitivity analyses. Firstly, we analyzed the EWAS models with additional adjustment for BMI in 370 subjects who had available data. Secondly, we analyzed the EWAS models with additional adjustment for the top 5 genetic principal components in 378 subjects with genotype data. Finally, we replicated the EWAS models in the subset of 286 participants who had hypertension and/or diabetes.

### Genome annotation

Adult human kidney chromatin states were generated by training a 15-state model using chromHMM software to capture all the relevant interactions between chromatin marks[38,40]. Histone modification data for adult human kidney, including H3K4me1, H3K4me3, H3K27ac, H3K36me3, H3K9ac, and H3K9me3, were obtained through ChIP-seq from GEO (GSM670025, GSM621648, GSM621651, GSM772811, GSM1112806, GSM621634, and GSM621638). Chromatin states for 127 tissues or cell types were obtained from the Roadmap Epigenomics project(https://egg2.wustl.edu/roadmap/web_portal/meta.html). The enrichment of DMPs in each annotation category was assessed by comparing the mapped number to all tested methylation probes present on the EPIC arrays. The significance of differences was determined using Fisher's exact test, with a P-value threshold of <0.05. Transcription factor motif enrichment was performed using HOMER (v4.10.3)[66]. Quantification and plotting of histone modifications were done using Deeptools[67].

### Methylation quantitative trait loci (meQTL)

To determine the human kidney cis-methylation quantitative trait loci (cis-meQTL) for CpG sites that are under a strong genotype effect, we investigated the associations between single nucleotide polymorphisms (SNPs) within a ±1 Mb cis window of CpGs and identified fibrosis-DMPs using MatrixQTL (v.2.1.0) R package[68]. Genotype data from kidney samples have been prepared using either Axiom Tx SNP GWAS array or Affymetrix Axiom Biobank array[6]. In brief, quality control was performed using PLINK (v1.9)[69]. A total of 549,142 SNP around these CpG sites (constituting 673,614 SNP–CpG pairs) were tested with an additive linear model fitted with covariates including general variables (sample collection site, age, sex, top five genotype PCs, degree of bisulfite conversion, sample plate and sentrix position) and PEER factors[6,70]. Significant cis-meQTLs were identified using an FDR threshold of <0.01.

## Single-nucleus RNA-seq profiles

Following manufacturer's protocol the Chromium Controller (10X Genomics, PN-120223) was used to load 30,000 cells into a Chromium Next GEM chip G Single Cell Kit (10X Genomics, PN-1000120) to generate single cell gel beads in the emulsion (10X Genomics, PN-1000121). The cDNA and library were created using the Chromium Next GEM Single Cell 3′ GEM Kit v3.1 (10X Genomics, PN-1000121) and Single Index Kit T Set A (10X Genomics, PN-120262), respectively[50].

## Single-nucleus chromatin accessibility profiles and differentially accessible regions

Single nucleus ATAC-seq libraries were generated using the Chromium Single Cell ATAC Library & Gel Bead Kit and Chromium i7 Multiplex Kit N Set A (10X Genomics, PN-1000084) according to the manufacturer's instructions as described earlier[50].

The Signac "FindMarkers" function was used to evaluate peaks observed in at least 20% of cells for differentially accessible chromatin regions (DARs) between different cell types using a likelihood ratio test, a log-fold-change threshold of 0.25, and an FDR of 0.05. ChIP-Seeker (v1.24.0) was then utilized to annotate the genomic regions harboring snATAC-seq peaks[71].

## Gene expression profiles

RNA-seq generation and pre-processing in kidney tissue have been previously described[6]. Briefly, RNA was isolated from the tubular compartment using the RNeasy mini-kit following the manufacturer's instructions. Transcript level changes were profiled by RNA-seq with reads aligned to the human genome (hg19) using STAR (v.2.4.1d)[72]. Gene expression level was quantified using RSEM (v.1.3.1) and further normalized across samples using edgeR (v3.32.1)[73,74]. The associations between gene expression and DNA methylation or clinical variables were evaluated using Pearson correlation analysis.

## Methylation risk score (MRS)

As many of the fibrosis-DMPs were correlated with each other, we performed penalized regression to narrow down candidate CpGs. Specifically, we applied the least absolute shrinkage and selection operator (LASSO) algorithm using the R package glmnet (v 4.1-7). Since previous kidney EWASs were conducted using the Illumina 450 K array and relatively small sample size, we randomly split the study samples into a 70% training set ($n = 279$) and a 30% testing set ($n = 120$). The penalty parameter λ was optimized using 10-fold cross-validation and the LASSO model was adjusted for the same main EWAS covariates that were not subject to penalization in the training set. We then integrated the LASSO-selected CpGs in the remaining testing set by calculating a weighted methylation risk score (MRS), which was the sum of the residualized methylation M values multiplied by the LASSO estimate effect size of each probe.

To discuss the clinical application of methylation information, we performed three separate analyses. 1) First, we investigate whether our established MRS could provide complementary information to kidney fibrosis estimations compared to the eGFR-based prediction model. We generated fibrosis estimates using a polynomial linear regression model, with eGFR as the predictor variable, controlling for age, sex, race, diabetes mellitus, and hypertension status[60]. We compared the model fitness between the model using the above clinical information alone and the model with additional MRS information by the likelihood ratio χ2 test. The 30/70 data split method was only used for independent testing of the fibrosis estimation analysis. Eventually, the weights for MRS were fine-tuned using all available data in 399 samples, and the updated MRS was applied to all samples for statistical inference. 2) To further evaluate the ability of MRS to independently stratify CKD risk, we conducted a logistic regression analysis, controlling for age, sex, race, diabetes mellitus, and hypertension status. We performed the receiver operating characteristic (ROC) curve analysis to quantify the discriminative performance of models using clinical information alone and with additional MRS. We used R package ROCR (v 1.0-11) to estimate 95% confidence intervals (CI) of the area under the ROC curves (AUC) and compared two models by the likelihood ratio χ2 test. 3) Lastly, we examined whether MRS could predict future kidney function decline in our cohort. We included 117 participants with at least 3 months of follow-up data after kidney biopsy and used a mixed-effect model to determine factors associated with longitudinal eGFR levels[75]. The mixed-effect model incorporated random-effects terms to account for patient and patient × time differences in the eGFR follow-up levels, controlling for known clinical covariates, including age, sex, race, BMI, diabetes mellitus, and hypertension status, using the R package lme4 (v 1.1.32).

## Methylation differences predict longitudinal eGFR change

The association of epigenome-wide methylation alterations and kidney disease development was investigated in the subset of 117 participants with longitudinal eGFR data available. The linear mixed model was used to characterize individual trajectories of longitudinal eGFR change. Random-effects terms were incorporated to account for patient and patient × time differences in the eGFR follow-up levels. The predictor included the term of methylation value multiplied by the time interval, while controlling for the same covariates as the main EWAS models. The EWAS analysis was conducted using the R package lme4 (v 1.1-32) and the coefficients were extracted using the R package broom (v 1.0.4). The epigenome-wide significance threshold ($P < 9.42 \times 10^{-8}$) was used to determine the significant CpGs.

## Reporting summary

Further information on research design is available in the Nature Portfolio Reporting Summary linked to this article.

## Data availability

The human kidney snATAC-seq data have been deposited with the Gene Expression Omnibus (GEO) under accession code nos. GSE172008, GSE200547 and the Common Metabolic Diseases Genome Atlas (https://cmdga.org/search/?type=Experiment&searchTerm=FNIH0000000). The Integrative Genomics Viewer visualization of human kidney cell-specific differentially accessible chromatin data generated in this study are provided at (https://susztaklab.com/Kidney_meQTL/index.php). Methylation data generated in this study have been deposited in Supplementary data 1 and Supplementary data 2. The raw individual participant data included in this project are protected and are not available due to data privacy laws. The human kidney eQTLs used in the present study are available online at the Susztaklab Kidney Biobank (https://susztaklab.com/Kidney_eQTL). The human kidney bulk RNA-seq data used in this study are available at GEO under accession numbers GSE115098 and GSE173343. The human kidney single cell ATAC-seq data are available in GEO under accession number GSE211785.

## Code availability

Customized code used in the present study are presented on GitHub (https://github.com/Yyan-med/Kidney_epigenomics) and Zenodo[76].

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

## Acknowledgements

Work in the Susztak lab is supported by NIH (grant nos. R01DK087635, R01DK076077 and R01DK105821 to K.S.). We thank the Molecular Pathology and Imaging Core (P30-DK050306) and Diabetes Research Center (P30-DK19525) at University of Pennsylvania for their services. The authors acknowledge the use of Biorender.com in creating schematic Fig. 3a.

## Author contributions

Y.Y. and K.S. conceived and designed the study. H.B.L. and X.S. contributed to the preparation and analysis of the EPIC data. A.A. was responsible for the preparation of the snATAC-seq and snRNA-seq libraries. M.P. conducted the histological scoring for the kidney tissues. Y.Y., K.S., and H.Z.L. performed the statistical analysis. Y.Y. and K.S. wrote the manuscript. K.S. coordinated and oversaw the study. All authors critically reviewed and discussed the results, and all authors edited and approved of the final manuscript.

## Competing interests

K. Susztak declares the following interests: research support from AstraZeneca, Bayer, Boehringer Ingelheim, Calico, Genentech, Gilead, GSK, Jnana, Lilly, Maze, Merck, Novartis, Novo Nordisk, Regeneron, Variant Bio, and Ventus; advisory board membership with Jnana Therapeutics and Pfizer; consultancy for AstraZeneca, Bayer, GSK, Jnana Therapeutics, Maze, Novo Nordisk, Pfizer, and Ventus; ownership of patents related to Jag1- and Notch-based targeting of chronic kidney disease; editorial board membership with *Cell Metabolism*, *eBioMedicine*, *Journal of the American Society of Nephrology*, *Journal of Clinical Investigation*, *Kidney International*, and *Med*. The remaining authors declare no competing interests.
