## [Peer Review File · Nature Communications]

Unraveling the Epigenetic Code: Human Kidney DNA Methylation and Chromatin Dynamics in Renal Disease DevelopmentREVIEWER COMMENTS

Reviewer #1 (Remarks to the Author):

Yan et al. shed light on the role of epigenetic changes in kidney disease development. The authors identify disease-associated differentially methylated positions influenced by genetic variations. The study also reveals regions showing both methylation and open chromatin changes, impacting genes expressed in proximal tubules and metabolism-associated genes. Additionally, the use of methylation risk scores (MRS) improves disease state annotation and prediction. Overall, this study provides valuable insights into the relationship between epigenetic changes and kidney disease, opening avenues for novel risk stratification methods. This impressive study was conducted with great expertise. Some suggestions for enhancing it further are below:

Major:

- The linear mixed-effects models include lymphocytic infiltration as one of the covariables. In extension of this, have you considered the variation in cell type mixtures in the tubule samples? In the analysis of blood samples this is one of our major covariate concerns. Could a reference free cell type estimation approach be applied to the tissue level DNAm data?
- What was the rationale between the two-staged linear mixed-effects + linear regression approach?
- eGFR was estimated with the 2009 CKD-EPI equation. In context of the mixed ancestry in the cohort it would be beneficial to use the 2021 race-free CKD-EPI equation. Given the many analyses depending on eGFR a sensitivity analysis of some of the main analyses such as the MRS ~ CKD associations might already be helpful.
- Does Supplementary Figure S5 include all CpGs? In the sense of a sensitivity analysis, it might be helpful to restrict this to the significant or the suggestive DMPs from the main analysis. This would also allow for the inclusion of the proportion of direction consistent effects in the text.
- As you are using Harmony to remove batch effects anyhow and the RNA analyses are self-contained (do not directly depend on e.g. the genetic data), could the snRNA seq dataset be integrated with the public domain data from KPMP (Lake et al.)? This should increase sample size considerable and make analyses even more robust and specific.
- Were MRS fibrosis models developed using un-transformed fibrosis scores? I wasn't sure based on the methods. Given the distribution of scores it seems like log2 transformed scores (as in the EWAS) would be beneficial for MRS development.
- Why was the design of MRS evaluation changed between fibrosis and CKD? Keeping the 70-30 split for CKD as well would simplify the design and provide stronger conclusions.
- MRS associations with incident CKD and eGFR decline should include baseline eGFR and ACR in the base models to see if the MRS can improve on top of these.
- Enrichment analyses did not account for multiple testing ($P < 0.05$). The number of tested annotation categories should be incorporated.

Minor:

- The specific manuscript proposing this would be a helpful reference: "An environmentally programmed transgenerational inheritance of kidney function has been proposed."
- "Several studies already explored epigenetic changes in patients with kidney disease. Cytosine methylation changes in blood samples have been analyzed in several large cohorts, including the Pima, CKDGen, DCCT (Diabetes Control and Complications Trial), and in the Chronic Renal Insufficiency Cohort (CRIC) 14,15,25-28." Current ref 16 includes blood DNAm in CKD patients (GCKD study).
- Are the 34 CpGs at the more stringent 1% Bonferroni level discussed or follow up on? Is there a specific enrichment or other difference to the 5% sites? If this was not investigated, it might be simpler for readers to drop the secondary definition of hits. Similarly for the 3 eGFR DMPs at 1%.
- If I am not mistaken - the results section on the two replication studies cites two different papers for the replication dataset 1 and non for dataset 2. Are both replication datasets 450k (as the text indicates)? If so, why are all 85 sites available in the second but not the first replication dataset? Is

there any overlap in individuals that were measured with the 450k and used for replication and the discovery EPIC cohort?

- Was mCpGs introduced? Or is this a typo "Significant cis-meQTL SNPs were mostly found within 100 kb of target DMPs (Fig. 3c), and those mCpGs were..."
- Add reference? "OAF (Out At First Homolog), a protein-coding gene expressed in the renal cortex, has been reported to be involved in protein reabsorption in the kidney."
- Is Figure 4a a reproduction of Figure 1e in Abedini et al. bioRxiv / Ref 48? Similarly for Figure 5b and Fig. 1c. Please indicate.
- Why is there no nearest gene for some DMPs? "Among the 171 fibrosis DMPs, 137 DMPs were annotated to 125 protein-coding genes"
- As diagnostics are repeatedly featured as a benefit of the MRS it seems warranted to discuss the requirement of a kidney biopsy to estimate the presented MRS.
- Future studies might benefit from the imputation to the TOPMed imputation panel instead of 1,000 Genome Phase 3.

Reviewer #2 (Remarks to the Author):

In the present article, Yan et al present results which highlight the associations between kidney epigenetic changes, eGFR and fibrosis; exploring the potential causal links between epigenetic changes and disease development. They do this by exploring the genetic variants which are associated with methylation and gene expression. This is an interesting and impressive analysis which I believe will be of interest to the kidney and epigenetic fields. However, I believe some more formal and detailed causal inference analysis would be beneficial.

The research team have made remarkable progress and contributions to the kidney epigenetic field in recent years and have numerous solid publications which improve our understanding of the role of epigenetic and genetic mediators of kidney disease. It would help to clarify how the present kidney meQTL analysis add to those previously published by the group (eg Liu et al. Nat Genet. 2022 performed DNA methylation profiling in 506 human kidneys and meQTL analysis in 443 human kidney samples and Sheng et al Nat. Genet. 2021). There also appear to be very relevant literature from other groups which is not discussed (EG: Breeze et al 2021 Genome med and Eales et al 2021 Nature Genetics).

It would help to clearly what this new study adds to existing ones.
What is the overlap in samples, findings, and analysis?

Another major weakness of this manuscript given the impressive resource available in the research team is the lack of experimental validation. The authors discuss that future studies should include experimental validation to fully understand mechanisms. I appreciate it is not feasible to perform detailed mechanistic studies for every epigenetic change identified, but the paper would benefit from some level of experimental validation; for example in demonstrating changes methylation/open chromatin/TF binding in the same samples: EG for a few of the key changes the authors highlight.

140: What is the overlap in CpGs identified in eGFR-EWAS and fibrosis-EWAS. Is there a difference in predicted functionality (of the CpGs - eg based on location, or of the genes?)

169-171: How do validation cohorts compare to discovery? For example: Do they all use micro-dissected tubular cells?

215-217: Is this finding compared to all probes on the EPIC array?
Results in figure 2a seem show different numbers to those discussed in the text.

236-240: Is there any evidence that these transcription factors are mediating the effects of differential methylation in kidney fibrosis? Does altering methylation alter transcription factor binding, or vice versa? The authors select cg04659689 as an example. But – can they show that transcription factor binding is changed too? Would be helpful to show methylation and transcription factor binding changes in the same samples.

275-281: What is the purpose of associating those SNPs with blood pressure? And 294-296: Overlap in genetic variants is not sufficient to “understand whether the underlying genetic variation-driven methylation changes drive disease development”. Further causal inference analysis is needed here.

351-357: How do these TFs compare with the ones identified in methylation analysis on dissected tubule? Also, DKD is mentioned here but (336-337) samples are control or CKD.

368-370 I am not clear which data shows the genetic variants modifying open chromatin?

434-435: The authors state “Interestingly, methylation differences were more common around genes expressed in proximal tubule (PT) cells” and 614-616: “Notably, the majority of the prioritized genes were found to be enriched in proximal tubule cells, supporting the notion that DNA methylation alterations may preferentially impact gene expression in these cells”
Yet, the study analyses changes in microdissected human kidney tubule samples (since the epigenome is cell type specific). Does this not explain why tubular cell genes are predominantly identified through the analysis pipeline?

463: “Fibrosis-associated DMPs affect the expression of solute carriers and metabolic genes” The authors need to provide more analysis (EG detailed causal inference) or perhaps some mechanistic study here to make this statement true. At present, the authors only show (albeit convincing) associations rather than direct effects of DMPs.

The samples used are taken from individuals with renal neoplasia. Since epigenetic changes (including methylation) play a major role in cancer development and progression, can the authors provide evidence that the changes they observe are not confounded by renal neoplasia? Does the ‘unaffected’ tissue from which these samples are collected truly represent normal tissue?
I would be interested to know if this has been formally tested (by this group or others)

The development of methylation risk scores is a novel and a very important analysis in the manuscript, and should be expanded on. For instance; can the authors also investigate the methylation risk score in blood as well as kidney tissue. This is an important point for the identification of prognostic biomarkers. Given the cell-specificity of epigenetic changes, it would be important to investigate this. How many of the cpgs in the methylation risk scores have robust mSNPs which could be tested?

Can the authors also validate the MRS in independent cohorts?

Methodology: more details of lymphocytic infiltration adjustment needed. How robust are those methods?

It would be helpful if the authors provide more information on ‘fibrosis score’ used. what proportion of the kidneys used in EWAS had fibrosis?

Minor points:

As mentioned: It would be helpful if the authors could clearly outline which are new data and the data which have been published previously, to emphasise what this new study adds to existing literature.

In the introduction: It is not clear why "metabolic memory" (or indeed, the sensitivity of Epigenome-modifying enzymes to metabolites) is mentioned: this is not a focus of the present study whose analysis predominantly focus on genetic influence on methylation. (likewise 514-515; I am not convinced non-genetic factors (environmental) have been "emphasised" as authors indicate)

Lines 77-78: The transgenerational nature of epigenetic changes is still debated. The authors should provide a reference to indicate that epigenetic changes are maintained through generations.

Reviewer Comments:

Reviewer 1

1. The linear mixed-effects models include lymphocytic infiltration as one of the covariables. In extension of this, have you considered the variation in cell type mixtures in the tubule samples? In the analysis of blood samples this is one of our major covariate concerns. Could a reference free cell type estimation approach be applied to the tissue level DNAm data?

Response: Thank you for your question. Acknowledging the importance of cell fraction changes we have incorporated lymphocytic infiltration as a covariate into our linear mixed-effects models. Lymphocyte infiltration was scored by our expert renal pathologist.

Reference methylation data is not available for human kidney cell types limiting our ability to estimate cell fractions in our methylation data. In our preliminary analysis, reference free cell proportion estimates generate less consistent results limiting its current applicability. We do acknowledge the importance cell type specific epigenetic changes and for this reason we included the snATACseq information.

2. What was the rational between the two-staged linear mixed-effects + linear regression approach?

Response: The motivation behind employing the two-staged linear mixed-effects and linear regression approach was to effectively account for confounders and covariates in the dataset in our analysis. This model enables us to appropriately account for the hierarchical nature of our data. We used a mixed effect model to account for technical variation, encompassing bisulfite conversion control, mean intensity of measurements, sample plate, lymphocytic infiltration, and batch-specific variability. Since these factors only affect the methylation levels, but not the outcome of interest (fibrosis score), we first applied the linear mixed-effects model to obtain adjusted methylation.

We then used the residualized methylation level in the downstream analysis to identify changes associated with the degree of fibrosis. In the linear regression analysis with the fibrosis score as the response, we adjusted for possible individual-level confounding variables.

Implementing this model empowered us to derive adjusted methylation levels for each individual taking into account the confounders listed above, mitigating potential confounding factors and enhancing the accuracy of subsequent linear regression analyses.

3. eGFR was estimated with the 2009 CKD-EPI equation. In context of the mixed ancestry in the cohort it would be beneficial to use the 2021 race-free CKD-EPI equation. Given the many analyses depending on eGFR a sensitivity analysis of some of the main analyses such as the MRS ~ CKD associations might already be helpful.

Response: Thank you for raising this important point. We agree that the 2021 race-free CKD-EPI equation could improve the eGFR estimation. To address this concern, we conducted sensitivity analyses for the **MRS ~ CKD associations** using the 2021 race-free equation. We identified 110 CKD patients and conducted a logistic regression analysis, controlling for age, sex, race, diabetes mellitus, and hypertension status. Our MRS prediction remained consistent across both approaches. Our MRS was significantly associated with (2021 race-free eGFR) CKD incidence (OR = 2.61, 95% CI = 1.37 – 5.05; $P = 3.7 \times 10^{-3}$). Compared to the initial model that only included age, sex, race, presence of DM, and/or HTN, the addition of MRS significantly improved the discriminative performance of incident CKD (area under the curve, 0.74 versus 0.70, likelihood ratio χ^2 test $P = 1.4 \times 10^{-5}$). This reaffirms the robustness of our results and underscores the integrity of our study's conclusions.

4. Does Supplementary Figure S5 include all CpGs? In the sense of a sensitivity analysis, it might be helpful to restrict this to the significant or the suggestive DMPs from the main analysis. This would also allow for the inclusion of the proportion of direction consistent effects in the text.

Response: We appreciate your suggestion. Supplementary Figure S5 includes all CpGs probes in our analysis. In response to your question, we undertook a sensitivity analysis focusing only on the significant DMPs, which yielded findings consistent with our primary analysis.

However, it's important to emphasize that our initial analysis applied a stringent threshold, which aimed to ensure the specificity of the most significant probes. Additionally, during our subsequent biological functional enrichment analysis, we extended the threshold to include the top 1000 probes to enrich for more meaningful signals.

5. As you are using Harmony to remove batch effects anyhow and the RNA analyses are self-contained (do not directly depend on e.g. the genetic data), could the snRNA seq

dataset be integrated with the public domain data from KPMP (Lake et al.)? This should increase sample size considerable and make analyses even more robust and specific.

Response: Thank you for your insightful suggestion. We totally agree that integrating our snRNA seq dataset with the KPMP data do could increase our sample size, consequently enhancing the statistical power and generalizability of our results. However, integrating datasets that are generated by different groups under different conditions are not easy, so we plan to integrate our results with the KPMP dataset in future analysis. Here we would like to share data that was generated in our own lab.

6. Were MRS fibrosis models developed using un-transformed fibrosis scores? I wasn't sure based on the methods. Given the distribution of scores it seems like log2 transformed scores (as in the EWAS) would be beneficial for MRS development.

Response: We apologize for any confusion this may have caused. In our primary analysis, we utilized un-transformed fibrosis scores to evaluate whether the addition of the established MRS could enhance the explanatory power of the natural distribution of fibrosis scores.

For the revised manuscript we repeated the analysis using log2-transformed fibrosis scores. The outcomes remained consistent. We used log2-transformed fibrosis score and a polynomial linear regression model, with eGFR as the predictor variable, controlling for age, sex, race, diabetes mellitus, and hypertension status. The eGFR-based prediction model explained 30.1% of the log2-transformed fibrosis score variance. The addition of MRS into the base model significantly improved this capability to 42.1% (likelihood ratio χ^2 test $P = 6.3 \times 10^{-6}$). This reaffirms that our established MRS can indeed enhance the accuracy of fibrosis estimation.

7. Why was the design of MRS evaluation changed between fibrosis and CKD? Keeping the 70-30 split for CKD as well would simplify the design and provide stronger conclusions.

Response: Thank you for your thoughtful question.

In the case of the fibrosis models, our main goal was to establish a reliable MRS capable of accurately predicting fibrosis scores. The candidate CpGs were selected from the fibrosis EWAS results, and it was important to validate the MRS's performance in an independent cohort. We did not have an external human kidney methylation dataset. The adoption of a 70-30 training-testing split was chosen to ensure a balanced representation of data, thus promoting the generalizability of the MRS to new samples. This approach enabled us to evaluate how well the MRS performed on previously unseen data, offering insights into its real-world applicability.

We did not build a separate risk prediction model for CKD, instead we tested whether the MRS score estimated based on the fibrosis score is associated with CKD and whether adding the MRS score can lead to improvement in CKD prediction. We would like to emphasize that CKD is defined by eGFR, not fibrosis score. Since the MRS scores are NOT defined for CKD, we feel that 70/30 training/testing splitting of the samples was not needed. Our analysis aimed to show the utility of the MRS defined using the fibrosis score in predicting CKD. By using all available samples, we could increase the sample size, allowing us to account for a broader array of factors that might influence CKD development.

8. MRS associations with incident CKD and eGFR decline should include baseline eGFR and ACR in the base models to see if the MRS can improve on top of these.

Response: We truly appreciate your insightful suggestion.

We utilized mixed-effect models to estimate the eGFR decline, which inherently account for baseline eGFR. This approach helps ensure the comprehensive assessment of the associations within the context of our study design.

Unfortunately, ACR data is not available for all of our participants. Limiting the sample size for subjects with ACR data would markedly reduce the statistical power or leading to possible bias in estimates if the missing is not at random. We included this issue as an important limitation of the presented work. "We were limited by the availability of albumin-to-creatinine ratio (ACR) data, which was only available for fewer than one-third of the participants. As a result, we were not able

to include ACR information into the CKD incidence and longitudinal eGFR decline prediction models.”

9. Enrichment analyses did not account for multiple testing ($P < 0.05$). The number of tested annotation categories should be incorporated.

Response: Thank you for highlighting this crucial aspect. Due to the stringent threshold we applied, the number of identified DMPs is small therefore the enrichment analysis significance diminishes following multiple testing adjustments. In response, we repeated the enrichment analysis by expanding the data for the top 1000 CpGs as DMPs, which allowed us to test for enrichment.

10. The specific manuscript proposing this would be a helpful reference: “An environmentally programmed transgenerational inheritance of kidney function has been proposed.”

Response: Thank you for bringing this to our attention. We carefully reviewed the literature and came to the consensus that the evidence supporting transgenerational inheritance in humans is indeed controversial and cannot be proven. We deleted the statement to avoid any potential confusion.

11. “Several studies already explored epigenetic changes in patients with kidney disease. Cytosine methylation changes in blood samples have been analyzed in several large cohorts, including the Pima, CKDGen, DCCT (Diabetes Control and Complications Trial), and in the Chronic Renal Insufficiency Cohort (CRIC) 14,15,25-28.” Current ref 16 includes blood DNAm in CKD patients (GCKD study).

Response: Thank you we included reference 16 here again.

12. Are the 34 CpGs at the more stringent 1% Bonferroni level discussed or follow up on? Is there a specific enrichment or other difference to the 5% sites? If this was not investigated, it might be simpler for readers to drop the secondary definition of hits. Similarly for the 3 eGFR DMPs at 1%.

Response: Thank you. As suggested, we dropped 1% Bonferroni corrected secondary analysis.

13. If I am not mistaken - the results section on the two replication studies cites two different papers for the replication dataset 1 and non for dataset 2. Are both replication

datasets 450k (as the text indicates?)? If so, why are all 85 sites available in the second but not the first replication dataset? Is there any overlap in individuals that were measured with the 450k and used for replication and the discovery EPIC cohort?

Response: Thank you for bringing up these points. Both replication datasets are indeed based on the 450k platform, and the discrepancy you noted is due to the distinct methodologies employed in these studies. In the first replication study led by Gluck, 321,473 probes were used and 65 significant CpGs were identified through a similar EWAS study design (fibrosis as continuous outcome, controlling for age, sex, race, DM, and HTN status, degree of lymphocytic infiltrate on histology and batch effect), analyzing 91 human kidney samples. Among our 171 fibrosis-DMPs, 55 were found to overlap with Gluck's dataset (which contained 321,473 probes), and all 55 CpG sites validated our primary analysis.

For the second replication dataset led by Yi-an, differential methylation regions were identified using a distinct approach that directly compared the methylation ratio between control and diseased tubule samples (binary outcome), and identified those with more than 50% difference in their methylation ratio and a P -value <0.01 as differentially methylated regions. Hence, we couldn't directly compare our results with Yi-An's dataset. We re-run the linear regression model in the GSE50847 dataset (485,577 probes) accordingly. Among our 171 fibrosis-DMPs, 85 CpG sites were present in this dataset, so we validated the 85 overlapped CpG sites here. We have provided a detailed explanation in the revised manuscript to clarify this issue.

14. Was mCpGs introduced? Or is this a typo "Significant cis-meQTL SNPs were mostly found within 100 kb of target DMPs (Fig. 3c), and those mCpGs were..."

Response: Thank you for pointing out this issue. The term "mCpGs" in that context refers to methylated CpGs. However, to avoid any misunderstanding, we have chosen to use "DMPs" (differentially methylated positions) instead of "mCpGs."

15. Add reference? "OAF (Out At First Homolog), a protein-coding gene expressed in the renal cortex, has been reported to be involved in protein reabsorption in the kidney."

Response: Thank you for your suggestion. We have included the reference to the mentioned gene in our manuscript.

16. Is Figure 4a a reproduction of Figure 1e in Abedini et al. bioRxiv / Ref 48? Similarly for Figure 5b and Fig. 1c. Please indicate.

Response: Thank you for pointing out this observation. We analyzed the dataset as presented in the bioRxiv paper for this figure. Figure 4a in our manuscript is a reproduction of Figure 1e from Abedini et al.'s bioRxiv manuscript (Ref. 48). The bioRxiv manuscript does not include analysis on differential accessibility regions (DAR). Furthermore, the final version of Abedini et al.'s study utilized a different and updated dataset.

17. Why is there no nearest gene for some DMPs? “Among the 171 fibrosis DMPs, 137 DMPs were annotated to 125 protein-coding genes“

Response: Thank you for bringing up this question. A gene is annotated as the nearest gene if the gene is within 5 kb. Most DMPs are on gene regulatory regions, many are on promoters but some are on enhancer regions that are not close to coding genes. While the majority of the 171 fibrosis DMPs were successfully annotated to protein-coding genes, a subset of DMPs may not have a nearest gene that meets our annotation criteria. This is an interesting observation, and we will further investigate it in the future.

18. As diagnostics are repeatedly featured as a benefit of the MRS it seems warranted to discuss the requirement of a kidney biopsy to estimate the presented MRS.

Response: Thank you for your observation. You are right; the potential need for a kidney biopsy to estimate the presented MRS is an important aspect to consider, especially given the repeated emphasis on diagnostics. We addressed this requirement and its implications in our discussion to provide a comprehensive view of the MRS's practical applications and limitations. “It's worth noting our study's reliance on kidney biopsy data is integral to the development and application of the MRS. While the MRS holds great promise in enhancing the precision of kidney disease diagnosis and prediction, it is important to recognize that obtaining kidney biopsy samples may not always be feasible or suitable for all individuals. Therefore, future research efforts in non-invasive (such as blood) alternatives for obtaining the data are needed to expand the accessibility and utility of the MRS in clinical practice.”

19. Future studies might benefit from the imputation to the TOPMed imputation panel instead of 1,000 Genome Phase 3.

Response: Thank you for your valuable suggestion. We agree that future studies could potentially benefit from using the TOPMed imputation panel instead of the 1,000 Genomes Phase 3 panel. The TOPMed panel offers a more comprehensive and updated reference for imputation, which could improve the accuracy and reliability of genetic imputation in our analyses.

Reviewer 2:

1. The research team have made remarkable progress and contributions to the kidney epigenetic field in recent years and have numerous solid publications which improve our understanding of the role of epigenetic and genetic mediators of kidney disease. It would help to clarify how the present kidney meQTL analysis add to those previously published by the group (eg Liu et al. Nat Genet. 2022 performed DNA methylation profiling in 506 human kidneys and meQTL analysis in 443 human kidney samples and Sheng et al Nat. Genet. 2021). There also appear to be very relevant literature from other groups which is not discussed (EG: Breeze et al 2021 Genome med and Eales et al 2021 Nature Genetics). It would help to clearly what this new study adds to existing ones. What is the overlap in samples, findings, and analysis?

Response: Thank you for your positive feedback on our team's contributions to the field of kidney epigenetics. In comparison to our previously published studies, such as Liu et al. (Nat Genet. 2022) and Sheng et al. (Nat Genet. 2021), the current study focuses on analyzing changes in the kidney epigenome in kidney disease. In Liu's paper, we mainly on kidney function GWAS loci identification and prioritized disease-causing genes, cell types, and regulatory circuits for 576 loci using multi-staged omics for GWAS annotation. The study used the same human kidney methylation arrays to calculate human kidney methylation QTL. The current work is focused on performing an EWAS analysis and generating MRS scores, which is distinct from the Liu et al publication.

We have taken a close look at these noteworthy papers. Breeze et al. used blood samples and concentrated on eGFR as the outcome, whereas our study uniquely utilized tissue samples that were not accessible elsewhere. Eales et al. also analyzed kidney tissues and examined the effect of genetic variants on methylation and gene expression and their critical relevance to the genetic regulation of blood pressure. We have cited these papers in the revised manuscript.

2. Another major weakness of this manuscript given the impressive resource available in the research team is the lack of experimental validation. The authors discuss that future studies should include experimental validation to fully understand mechanisms. I appreciate it is not feasible to perform detailed mechanistic studies for every epigenetic change identified, but the paper would benefit from some level of experimental validation;

for example in demonstrating changes methylation/open chromatin/TF binding in the same samples: EG for a few of the key changes the authors highlight.

Response: We appreciate your thorough evaluation of our manuscript and your valuable insights on the potential areas for improvement. In the current study, we aimed to identify DMPs associated with kidney traits and explore their manifestation, including genome location, cell specificity, as well as potential target genes. We focused on validation using external methylation datasets and external open chromatin datasets. Experimental validation can also enhance the robustness and impact of our findings, however at this point meaningful editing the methylation of a single methylated position is exceedingly difficult if not impossible. Another critical problem is that we would need human kidney cells with the methylation profile of the cells in vivo in the kidney. It is not possible to culture most human kidney cells (only PT and podocytes) and the methylation profile of cells in culture are very different from cells in the body. These are not trivial experiments requiring considerable time and effort and we hope to perform them in the future.

3. 140: What is the overlap in CpGs identified in eGFR-EWAS and fibrosis-EWAS. Is there a difference in predicted functionality (of the CpGs - eg based on location, or of the genes?)

Response: Five of the DMPs overlapped between the eGFR-EWAS and fibrosis-EWAS datasets. These overlapping CpGs are: cg01453157, cg04659689, cg25544164, cg27083891, and cg27630540.

In terms of predicted functionality, we observed small differences between the CpGs associated with eGFR and fibrosis. Specifically, the fibrosis-associated DMPs were likely to be enriched in biological processes related to metabolism, cell proliferation, apoptosis, and tubulointerstitial inflammation. On the other hand, the eGFR-associated DMPs were likely to be enriched in processes related to urinary protein resorption and ion transportation.

4. 169-171: How do validation cohorts compare to discovery? For example: Do they all use micro-dissected tubular cells?

Response: Thank you. Our validation cohorts also utilized micro-dissected tubular cells to align with the methodology used in the discovery cohort.

5.215-217: Is this finding compared to all probes on the EPIC array? Results in figure 2a seem show different numbers to those discussed in the text.

Response: We greatly appreciate your comment. We apologize for the typo and we have corrected it accordingly. Furthermore, we have conducted a thorough review of the manuscript to ensure the accuracy of all other data and text.

6. 236-240: Is there any evidence that these transcription factors are mediating the effects of differential methylation in kidney fibrosis? Does altering methylation alter transcription factor binding, or vice versa? The authors select cg04659689 as an example. But – can they show that transcription factor binding is changed too? Would be helpful to show methylation and transcription factor binding changes in the same samples.

Response: We sincerely appreciate your insightful questions.

Compared to all probes on the EPIC array, fibrosis-DMPs were enriched for 15 transcription factor binding motifs (false discovery rate (FDR) < 0.05; Fig. 2d, Supplementary Table S7), with the most prominent motifs representing ERRA, COUP-TFII, NUR77, and FXR, HNF4A, PPARA, HNF1B.

DNA methylation physically alters transcription factor-binding affinity, primarily through the methylated DNA-binding proteins that block transcription factor binding (Parry L et al., *Genes Cancer*, 2011(2); Björn Tampe et al., *Nephrology Dialysis Transplantation*, 2014(29)). In our prior investigations, we noted a higher prevalence of fibrosis-associated differential methylation within regions where transcription factors commonly bind. We inferred that these regions might have greater functional significance, given their potential to modify transcriptional accessibility. To elucidate the relationship between methylation and transcript level changes, our earlier study (Yi-An Ko et al., *Genome Biology* 2013(14)) analyzed gene expression and cytosine methylation patterns in tubule epithelial cells both before and after a 9-day treatment with the DNA methyltransferase inhibitor, decitabine. After decitabine treatment, we observed a substantial number of loci with concurrent differential methylation and gene expression changes, mirroring findings seen in CKD. This suggests that cytosine methylation changes in CKD may indeed be a driving force behind alterations in gene expression.

The expected location of probe cg04659689, is in the enhancer region of the OAF gene. Therefore, it may not be the most suitable for illustrating transcription factor binding. Instead, we present data for probe cg09837037 (chr 5: 1201601-1201603), which is located in the SLC6A19 promoter

region. We observed a positive correlation between the methylation level of cg09837037 and fibrosis, and a negative correlation between methylation and SLC6A19 expression (Pearson correlation coefficient $r = -0.56$, $P < 2.2 \times 10^{-16}$). Additionally, SLC6A19 gene expression in human kidneys exhibited a significant correlation with fibrosis ($r = -0.53$, $P < 2.2 \times 10^{-16}$). Our analysis on the IGV website (<http://www.susztaklab.com/>) indicates that lower open chromatin levels in disease state, especially in the PT cells.

7. 275-281: What is the purpose of associating those SNPs with blood pressure? And 294-296: Overlap in genetic variants is not sufficient to “understand whether the underlying genetic variation-driven methylation changes drive disease development”. Further causal inference analysis is needed here.

Response: Regarding the association of SNPs with blood pressure, we apologize for any confusion that might have arisen. While blood pressure itself may not be the primary focus of our study, it's important to recognize that blood pressure plays an important role in kidney disease development. Therefore, we included information about the association of SNPs with blood pressure to provide potential context for the identified genetic variants and their potential relevance in kidney fibrosis.

Regarding the overlap analysis, we appreciate your point. It is undeniably true that the mere identification of overlapping genetic variants offers only a partial view of the broader picture concerning the potential contributions of genetic variation-triggered methylation changes to the development of diseases. In our study, our principal objective here was to explore the existence of genetic-driven CpG sites that may have associations with various diseases, with a particular emphasis on those closely linked to kidney disease, such as hypertension, diabetes, proteinuria, and elevated uric acid levels. We have subsequently revised our statement to acknowledge this limitation in our discussion.

8. 351-357: How do these TFs compare with the ones identified in methylation analysis on dissected tubule? Also, DKD is mentioned here but (336-337) samples are control or CKD.

Response: We greatly appreciate your insightful questions. The majority of the enriched TFs were similar, including HNF4A, RUNX1, and Erra, which were known to play an important role in kidney disease. The difference was that we emphasized the cell-type level motif enrichment, which provides a more specific understanding of the regulatory processes in kidney cells.

Thank you for pointing out our typo, we changed to CKD.

9. 368-370 I am not clear which data shows the genetic variants modifying open chromatin?

Response: Sorry for the confusion here. Our analysis did not directly demonstrate genetic variants modifying open chromatin. However, it is important to note that the DNA in the nucleus is densely packaged into chromatin, a DNA–protein complex. Genomic regions that are actively involved in gene regulation must be accessible to transcription factors and are therefore located in areas of open chromatin. We provided clarification.

10. 434-435: The authors state “Interestingly, methylation differences were more common around genes expressed in proximal tubule (PT) cells” and 614-616: “Notably, the majority of the prioritized genes were found to be enriched in proximal tubule cells, supporting the notion that DNA methylation alterations may preferentially impact gene expression in these cells”. Yet, the study analyses changes in microdissected human kidney tubule samples (since the epigenome is cell type specific). Does this not explain why tubular cell genes are predominantly identified through the analysis pipeline?

Response: Thank you for bringing this point to our attention. Indeed, our study has focused on analyzing microdissected samples that are enriched for proximal tubules, therefore these might be the expected results. We have incorporated a statement into the text.

11. 463: “Fibrosis-associated DMPs affect the expression of solute carriers and metabolic genes” The authors need to provide more analysis (EG detailed causal inference) or perhaps some mechanistic study here to make this statement true. At present, the authors only show (albeit convincing) associations rather than direct effects of DMPs.

Response: Thank you, we agree with you. The presented study shows the association between DMP and solute carrier expression. We cannot be certain that this is a causal association. We have adjusted the text.

12. The samples used are taken from individuals with renal neoplasia. Since epigenetic changes (including methylation) play a major role in cancer development and progression, can the authors provide evidence that the changes they observe are not confounded by renal neoplasia? Does the ‘unaffected’ tissue from which these samples are collected truly represent normal tissue? I would be interested to know if this has been formally tested (by this group or others)

Response: We appreciate your thoughtful inquiry. The kidney tissue samples utilized in our study were obtained from a non-neoplastic portion of the kidney, specifically from regions at least 2 centimeters away from cancer. Careful histological analysis was implemented to ensure that the tissue analyzed in our study truly represents non-cancerous tissue. Formal comparison between biopsy and nephrectomy samples have not been performed as it is not ethical to obtain kidney biopsies from healthy subjects. Gene expression data obtained from CKD biopsies and nephrectomies show high concordance.

13. The development of methylation risk scores is a novel and a very important analysis in the manuscript, and should be expanded on. For instance; can the authors also investigate the methylation risk score in blood as well as kidney tissue. This is an important point for the identification of prognostic biomarkers. Given the cell-specificity of epigenetic changes, it would be important to investigate this. How many of the cpgs in the methylation risk scores have robust mSNPs which could be tested? Can the authors also validate the MRS in independent cohorts?

Response: We greatly appreciate your thoughtful suggestions.

The MRS developed in our study was primarily based on DMPs identified in kidney tissues. It's crucial to note that epigenetic patterns in kidney tissues substantially differ from those in blood samples. Therefore, methylation information from kidney tissue is not transferrable to methylation data of blood samples. Considering tissue-specificity is vital when assessing the applicability of our MRS. We have validated our results using a 70-30 training-testing split setting, which enable us to evaluate how well the MRS performed on new samples. However, regrettably, at present, we do not have access to independent cohorts with human kidney EPIC data that would allow for such validation. We have included it as one of the limitations of our study.

14. Methodology: more details of lymphocytic infiltration adjustment needed. How robust are those methods?

Response: Thank you for your inquiry regarding the methodology. Immune and lymphocyte infiltration and cell heterogeneity plays important role methylation variation. Lymphocytic infiltration was scored in each sample by an expert renal pathologist. This is part of standard evaluation of kidney biopsy samples in our center. The score was included as a covariate within our linear mixed-effects models. In a separate experiment we found that the pathologist scored lymphocyte infiltration correlated with genes expressed by lymphocytes.

15. It would be helpful if the authors provide more information on 'fibrosis score' used. what proportion of the kidneys used in EWAS had fibrosis.

Response: Thank you. Fibrosis score was determined following the practice established in the clinical practice. Adjacent kidney samples fixed in 10% neutral formalin and embedded in paraffin. The prepared sections were staining with Hematoxylin and eosin and periodic acid schiff. The samples were examined and analyzed by expert renal pathologist. Our prior studies indicated a strong correlation between fibrosis scores and gene expression (Beckerman et al. eBioMedicine). We had fibrosis score for every sample. There is no clear threshold for fibrosis. 129 samples had fibrosis scores greater than 10%, which is often used in the clinical practice to define disease, however this threshold is not been prospectively evaluated.

16. As mentioned: It would be helpful if the authors could clearly outline which are new data and the data which have been published previously, to emphasise what this new study adds to existing literature.

Response: We appreciate the reviewer's suggestion. We have included text and appropriate citations. Just to recap, while the kidney methylation data has been used to generate kidney methylation QTL information, we have not examined the relationship between methylation and disease severity (EWAS). Similarly, the snATAC dataset has not been analyzed for changes in the disease state.

17. In the introduction: It is not clear why “metabolic memory” (or indeed, the sensitivity of Epigenome-modifying enzymes to metabolites) is mentioned: this is not a focus of the present study whose analysis predominantly focus on genetic influence on methylation. (likewise 514-515; I am not convinced non-genetic factors (environmental) have been “emphasised” as authors indicate)

Response: Thank you. In the current study, we analyzed changes in methylation and open chromatin in disease states. We tried to dissect the genotype from non-genotype (environmental) effects, as the genotype effect on methylation can be estimated in methylation QTL studies. Our results indicate that most changes are not explained by genetic differences, as defined in meQTL studies. While genetic variations undoubtedly play a role in shaping epigenetic patterns, our results demonstrate that these genetic differences alone do not account for the entirety of the observed methylation alterations. As we mentioned in the introduction section, epigenetic modifications could account for some of the missing heritability of kidney disease, including “metabolic memory” phenomena. Our findings underscore the complexity of the disease etiology, suggesting that multitude of factors beyond genetics must be considered to gain a more holistic perspective on disease pathogenesis. We have elaborated on these points in the discussion section.

18. Lines 77-78: The transgenerational nature of epigenetic changes is still debated. The authors should provide a reference to indicate that epigenetic changes are maintained through generations.

Response: Thank you. Here we simply stated the one definition for epigenetic. The evidence for transgenerational inheritance in humans is controversial and cannot be proven, therefore we deleted the statement.

REVIEWERS' COMMENTS

Reviewer #1 (Remarks to the Author):

Thank you very much for the excellent rebuttal that addresses all my comments and suggestions.

Reviewer #2 (Remarks to the Author):

I thank the authors for their considered response - I have no further comments.

Reviewer #1 (Remarks to the Author):

Thank you very much for the excellent rebuttal that addresses all my comments and suggestions.

Response: Thank you!

Reviewer #2 (Remarks to the Author):

I thank the authors for their considered response - I have no further comments.

Response: Thank you!